# An integrated Mg battery-powered iontophoresis patch for efficient and controllable transdermal drug delivery

Yan Zhou[1], Xiaoteng Jia [2] ✉, Daxin Pang[3], Shan Jiang[1], Meihua Zhu[1], Geyu Lu[2,4], Yaping Tian [5] ✉, Caiyun Wang [6] ✉, Danming Chao [1] ✉ & Gordon Wallace [6]

Wearable transdermal iontophoresis eliminating the need for external power sources offers advantages for patient-comfort when deploying epidermal diseases treatments. However, current self-powered iontophoresis based on energy harvesters is limited to support efficient therapeutic administration over the long-term operation, owing to the low and inconsistent energy supply. Here we propose a simplified wearable iontophoresis patch with a built-in Mg battery for efficient and controllable transdermal delivery. This system decreases the system complexity and form factors by using viologen-based hydrogels as an integrated drug reservoir and cathode material, eliminating the conventional interface impedance between the electrode and drug reservoir. The redox-active polyelectrolyte hydrogel offers a high energy density of 3.57 mWh cm$^{-2}$, and an optimal bioelectronic interface with ultra-soft nature and low tissue-interface impedance. The delivery dosage can be readily manipulated by tuning the viologen hydrogel and the iontophoresis stimulation mode. This iontophoresis patch demonstrates an effective treatment of an imiquimod-induced psoriasis mouse. Considering the advantages of being a reliable and efficient energy supply, simplified configuration, and optimal electrical skin-device interface, this battery-powered iontophoresis may provide a new non-invasive treatment for chronic epidermal diseases.

Transdermal drug delivery is a patient-friendly therapy that allows for a sustained release profile with reduced side effects[1–3]. To improve the delivery efficiency, efforts have evolved towards seeking external forces to energetically fuel the drug across the stratum corneum (10–20 μm thickness). Iontophoresis is a safe, efficacious, and non-invasive delivery technology with minimized irritation to the skin compared with other facilitated transdermal delivery strategies, such as microneedles[4,5], electroporation[6], and thermal ablation[7].

Iontophoresis relies on electromigration to drive charged drugs under a direct electric field. As such, a stable power source is indispensable for the operation. An external power supply places a physiological and psychological load on patients[8]. Thus, wearable or skin-mountable iontophoresis that incorporates lightweight and safe power sources is highly desirable for patient-comfort[9,10]. The intensively explored energy harvesting strategy currently offers new options for self-powered iontophoresis devices, yet with limitations. The nanogenerators that harvest energy from human motion are

[1]College of Chemistry, Jilin University, Changchun 130012, China. [2]State Key Laboratory of Integrated Optoelectronics, College of Electronic Science and Engineering, Jilin University, Changchun 130012, China. [3]College of Animal Sciences, Jilin University, Changchun 130062, China. [4]International Center of Future Science, Jilin University, Changchun 130012, China. [5]Department of Dermatology and Venerology, The First Hospital of Jilin University, Changchun 130021, China. [6]ARC Centre of Excellence for Electromaterials Science, Intelligent Polymer Research Institute, AIIM Facility, University of Wollongong, North Wollongong, NSW, Australia. ✉e-mail: xtjia@jlu.edu.cn; typ@jlu.edu.cn; caiyun@uow.edu.au; chaodanming@jlu.edu.cn

constrained by the low (nano- to micro-ampere level) and short (min-long) burst of energy output, resulting in low and inconsistent delivery efficacy[11–14]. The enzymatic biofuel cell with human biofluids (e.g., glucose, lactate, and ethanol in sweat) as energy sources[15,16], depends on enzymatic activity and sufficient supply of biofluids, which limit the long-term operation on the skin. Battery-free wireless harvesting of radiofrequency power is also challenging due to the low energy transfer efficiency and constrained proximity to the antenna[17,18]. Hence, most commercial iontophoresis patches still rely on primary batteries as the power source. A significant advantage of primary batteries lies in them being reliable, uninterruptable, and effective energy supplies that enable high-dosage administration for long periods of operation[19–21].

The basic design of self-powered iontophoresis consists of a power source and a drug reservoir facing the skin surface. The drug reservoir is critical to harmlessly facilitate transdermal delivery within a small window of currents/voltages. Hydrogels comprised of cross-linked porous polymer networks possess the advantages of softness and tunable drug-loading capacity, allowing for the facile transport of payload drugs. The commonly used self-powered iontophoresis couples the power source with hydrogel-based drug reservoirs[22–24], however, this configuration increases the complexity and form factors of the iontophoresis device. In this system, the electrode and hydrogel drug reservoir are separated, which requires a power management module and connection with bulky wires and conductive pastes[25], resulting in an additional interface between the drug reservoir and the electrode. The electron current has to be converted to ion current at the electrode/drug reservoir interface to induce the diffusion of therapeutic agents by charge repulsion[26]. This conversion can induce a local heating effect due to the electrode overpotential, thus decreasing the electrical stimulation efficiency.

To address these issues, we present a simplified wearable iontophoresis patch with a built-in Mg battery using cytocompatible viologen-based hydrogel as both a drug-reservoir and cathode material. This polyelectrolyte hydrogel P(AM-co-SV), copolymerized from acrylamide (AM, offering cross-linked elastic networks) and p-styrene-bipyridine (SV, offering high redox activity) monomers, provide a stable and redox-controlled response to electro-stimuli. Compared to the widely used conducting polymers for electro-stimulated drug release, this polyelectrolyte hydrogel outperforms conducting polymers by leveraging a low driven potential for drug release, anti-bacterial activity to prevent infection, as well as an optimal bioelectrical interface for efficient transdermal delivery when deployed in the epidermal patch. The integrated ion-conducting electrode ensures direct electrochemical reduction of viologen by the Mg electrode and expulsion of incorporated drugs, thereby avoiding the interface impedance and associated Joule heating in the conventional design of separate electrode and drug reservoir (Supplementary Fig. 1), realizing an improved delivery efficiency. This wearable Mg battery-powered iontophoresis demonstrates stable current outputs and enhanced release capacity compared with energy harvester strategies (Supplementary Table 1), providing new therapeutic approaches for chronic skin diseases requiring precise drug delivery.

## Results

### Design and properties of P(AM-co-SV) hydrogels as drug-loading electrodes

The drug-loading electrodes for iontophoresis are required to be soft for conformal contacts and low interface impedance for electrically triggered efficient delivery[27]. P(AM-co-SV) hydrogels are ideal for the bioelectronic interface due to their ultra-low modulus, intrinsic ion conduction, and high mass permeability. They were synthesized by the free radical polymerization of AM and SV monomers under a potassium persulfate (KPS)/N, N, N′, N′-tetramethylethylenediamine (TEMED) redox initiation system (Fig. 1a). To incorporate redox-active

viologen groups into the hydrogel network, we firstly synthesized p-styrene-bipyridine (SV) monomer via the Menshutkin reaction (Supplementary Fig. 2). The formed hydrogels with SV contents of 0%, 5%, 10%, and 20% (wt%) were prepared and named PAAm, VH5, VH10, and VH20, respectively (formulations and detailed properties are listed in Supplementary Table 2).

Dexamethasone sodium phosphate (Dex) was incorporated as counter-ions into the viologen matrix (quaternary ammonium cations in the oxidized form) via electrostatic interactions by soaking into the drug-containing phosphate-buffered saline (PBS) solution. Dex-loaded hydrogel showed a loose porous morphology at the equilibrium of swelling, with the pore size gradually increasing with viologen contents (Supplementary Fig. 3). The porous three-dimensional network of the interfacing hydrogel was beneficial for the facile diffusion and transport of drug molecules. The distribution of C, N, and P elements determined by energy dispersive X-ray spectroscopy (EDX) confirms that Dex is uniformly loaded in the interior of the hydrogel (Fig. 1b). P(AM-co-SV) hydrogels demonstrated similar maximum swelling ratios (Supplementary Fig. 4a) and significantly improved drug-loading capacity with viologen contents. VH20 had a maximum drug-loading capacity of 7.16 mg g$^{-1}$, which was 3.96 times higher than PAAm (Fig. 1c). This drug-loading capacity was higher than conductive hydrogel-based drug reservoirs due to the strong electrostatic interactions[28–30]. The formation of P(AM-co-SV) hydrogels was confirmed by Fourier transform infrared (FTIR) spectroscopy (Fig. 1d). The intensity of peaks at 3000 and 1600 cm$^{-1}$ associated with benzene ring stretching vibration became stronger with more viologen contents, indicating effective grafting of viologen pendants.

Figure 1e demonstrates the gelation transition process and stretching (158%) of VH20 hydrogels. The PAAm-based hydrogel was used in this work because of its inherent softness with a low elastic modulus of 0.4–1.2 kPa (lower than 10–100 kPa for human skin)[31]. It can form a stationary quasi-solid yet moist interface with human skin. The tensile strength of P(AM-co-SV) hydrogels gradually declined while the strain at break increased with more viologen contents (Fig. 1f). Continuously increasing the hydrophilic side-chain aggregation region slightly decreased the elastic modulus but improved the toughness compared with PAAm (Supplementary Fig. 4b). The cross-linking points of hydrogels would be successively broken during the stressing process, contributing to energy dissipation. Loading-unloading tests further evaluated the energy dissipation ability of VH20 at increasing levels of tensile strain (Supplementary Fig. 4c). The hysteresis loop became more prominent with increasing maximum strain, indicating that more covalent bonds were sacrificed and thus dissipated energy[32]. The hysteresis loop gradually decreased over 10 cycles, indicating poor fatigue resistance due to the covalent cross-linking of VH20 hydrogels (Supplementary Fig. 4d). The dynamic rheological behaviors of PAAm and VH20 hydrogels as viscoelastic materials were investigated by detecting storage modulus (G′) and loss modulus (G″). In the frequency range of 0.1–100, G′ was dominant in the frequency-dependent oscillatory shear rheology at a constant strain of 1% (Fig. 1g). Meanwhile, G′ and G″ of PAAm were consistently higher than those of VH20, indicating the significant cross-linking density and high strength of PAAm. In addition, G′ was linearly related to frequency, reflecting the covalent chemical cross-linking characteristics[33].

### Cytocompatibility and antibacterial properties of P(AM-co-SV) hydrogels

Good biocompatibility is the prerequisite for hydrogels used in the skin-device interface over long-period usage. PAAm, a synthetic hydrophilic polymer, has been reported as protein resistant and biocompatible[34]. However, serious concerns remain about the safety issues of viologen, considering its potential cytotoxicity incurred from its involvement in the organism redox processes. We first assessed the cytotoxicity of VH20 and PAAm hydrogel extracts with different

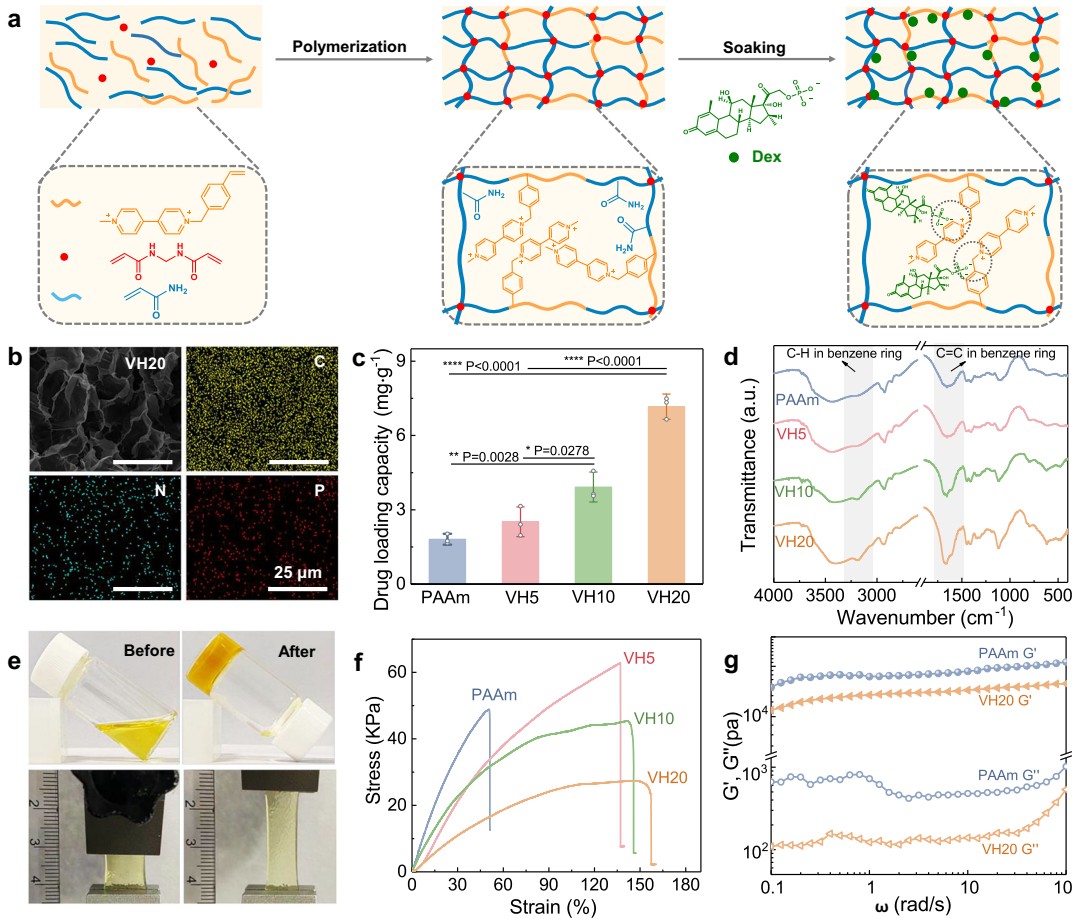

**Fig. 1 | Preparation and characterizations of P(AM-co-SV) hydrogels.**
**a** Schematic illustration of the synthesis, molecular structures, and interactions of drug-loaded P(AM-co-SV) hydrogels. **b** Scanning electron microscopy (SEM) and corresponding EDS images of the Dex-loaded VH20 hydrogel. Measurements were repeated three times independently with similar results. **c** Loading capacity of P(AM-co-SV) hydrogels (*n* = 3 independent experiments, data are presented as mean ± SD, *$P < 0.05$, **$P < 0.01$, ***$P < 0.001$, ****$P < 0.0001$, ns for no significance, *P* value was generated by one-way analysis of variance (ANOVA), followed by Tukey's multiple-comparison post hoc test). **d** FT-IR of P(AM-co-SV) hydrogels with different amounts of viologen. **e** Images of the gelation transition process and stretching of VH20 hydrogels. **f** Tensile curves of P(AM-co-SV) hydrogels with different amounts of viologen. **g** Angular frequency-dependent oscillatory rheology of PAAm and VH20 hydrogels.

concentrations on mouse fibroblast cells (L929 cells) by 3-[4,5-dimethylthiazol-2-yl]-2,5 diphenyl tetrazolium bromide (MTT) assay. Although the cell viability dropped as hydrogel extract concentrations increased, overall cell viability was always greater than 90% (Fig. 2a), indicating the low cytotoxicity of VH20. The proliferation curve of L929 cells cultured with VH20 extract (0.1 g mL$^{-1}$) was consistent with PAAm (Fig. 2b). The cell growth was slightly inhibited by the VH20 extracts, with reduced inhibitory impact at low hydrogel extraction concentrations (Supplementary Fig. 5).

We further demonstrated the cytocompatibility by culturing L929 cells on PAAm and VH20 hydrogels surfaces. Live/dead fluorescence assays showed that cells were spread well and displayed the regular spindle shape on the hydrogel surface (Fig. 2c). It showed good cell proliferation during the three-day culture, attributed to the hydrophilic nature of PAAm and porous network structure, providing adequate space and area for cell adhesion. The cytotoxicity reduction may be caused by the low dose of viologen (0.9 mg cm$^{-2}$) and covalently grafting of viologen pendants, which further hindered the release of viologen and their participation in redox processes in live organisms[35]. In addition, we verified the cytocompatibility when VH20 hydrogels were electrochemically reduced in a Mg battery. Cells could adhere and proliferate properly on the VH20 hydrogel surface when the battery was discharged for 30 min each day with various installed resistors (Supplementary Fig. 6).

An antibacterial property is advantageous for the skin-device interface because bacteria tend to adhere to traditional hydrogels, leading to a skin infection. P(AM-co-SV) hydrogels containing the quaternary ammonium salt in viologen may alter bacterial cytoplasmic membrane permeability, resulting in bacterial death caused by the blocked path of metabolism[36]. The antibacterial activity of VH20 hydrogel with different volumes was evaluated by contact sterilization experiments against *Escherichia coli* (*E. coli*, Gram-negative bacteria) and *Staphylococcus aureus* (*S. aureus*, Gram-positive bacteria). VH20 hydrogel showed significant volume-dependent antibacterial activity against both bacteria over 96 h compared to the control (Fig. 2d and e). In contrast to the cloudy control solution, the culture solution with 3 cm$^3$ of VH20 hydrogel was more transparent than the small volumes (Fig. 2f), consistent with the bacteria growth curves. The bactericidal rate and plate coating experiments demonstrated that VH20 hydrogel suppressed the growth of *E. coli* and *S. aureus*, and the inhibition of *S. aureus* was more evident (Supplementary Fig. 7). Taken together, P(AM-co-SV) hydrogels showed great potential as electrical skin interface due to their ultra-soft nature, cytocompatibility, and antibacterial capabilities.

## Electrochemical properties of P(AM-co-SV) hydrogels and Mg batteries

Sufficient ionic conductivity and low tissue impedance of drug-loading electrodes are needed for transdermal delivery[37]. As polyelectrolyte

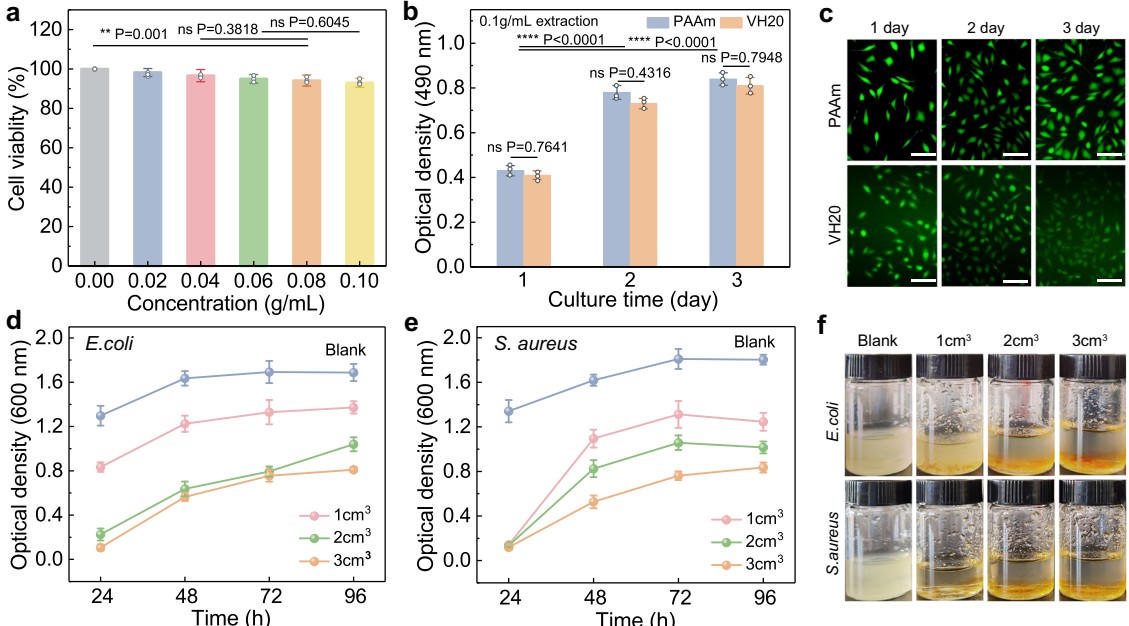

**Fig. 2 | Cytocompatibility and antibacterial properties of P(AM-co-SV) hydrogels. a** Cell viability of L929 cultured with various dosages of VH20 hydrogel extracts. ($n$ = 3 independent experiments). **b** L929 cell proliferation cultured with 0.1 g mL$^{-1}$ of VH20 hydrogel extracts and PAAm ($n$ = 3 independent experiments). **c** Fluorescence micrographs of L929 cultured on PAAm and VH20 hydrogels. **d** The optical density of VH20 hydrogels after co-culture with *E. coli* and (**e**) *S. aureus* ($n$ = 3 independent experiments). **f** Photographs of different volumes of VH20 hydrogel co-cultured with *E. coli* and *S. aureus* after 24 h. Data are presented as mean ± SD in (**a**, **b**, **d**, and **f**). *$P$ < 0.05, **$P$ < 0.01, ***$P$ < 0.001, ****$P$ < 0.0001, ns for no significance, $P$ value was generated by one-way analysis of variance (ANOVA), followed by Tukey's multiple-comparison post hoc test.

hydrogels with a fixed charge on the polymer network, P(AM-co-SV) are ideal for the bioelectronic interface. The ionic conductivity of P(AM-co-SV) hydrogels measured by electrochemical impedance spectroscopy (EIS) showed controllability ranging from 0.30 to 0.99 mS cm$^{-1}$ with more viologen contents (Supplementary Fig. 8a). This conductivity was on par with the PAAm-based ion-conducting hydrogels[38]. After loading with Dex, VH20 hydrogel showed improved ionic conductivity of 12.4 mS cm$^{-1}$, attributed to the free mobile ions in the water-swollen pores. Besides, the hydrogel conductivity was visualized when the brightness of the light-emitting diode (LED) bulb gradually dimmed as the effective distance increased (Supplementary Fig. 8b).

Cyclic voltammogram (CV) curves showed that peak currents of P(AM-co-SV) hydrogels gradually increased with the viologen content (Fig. 3a). Two pairs of distinct redox peaks at −0.39 V/−0.57 V and −0.80 V/−1.00 V were observed for VH20 hydrogel, which can be attributed to the transitions between dication (V$^{++}$), radical (V$^{+}$), and neutral (V$^{0}$) forms of the viologen segment[39]. The charge storage kinetics of VH20 hydrogel was also illustrated at various scan rates. The $b$ values from the linear fitting curves were from 0.53 to 0.81 at high scan rates, indicating a mixed surface and diffusion-controlled process (Supplementary Fig. 9). The Nyquist plots demonstrated that VH20 hydrogel showed the lowest bulk and charge transfer resistance (Fig. 3b), consistent with the ionic conductivity results. To evaluate the performance of the iontophoresis patch in vitro, a tissue interface impedance test was performed on the front of the forearm. The VH20 hydrogel electrode had a much lower impedance (4.3 × 10$^5$ Ω) than PAAm (3.2 × 10$^6$ Ω) at a low frequency of 1 Hz (Fig. 3c), which enabled precise transdermal performance. The impedance-lowering effect of viologen was more prominent at 1 Hz (low frequency) than 10$^2$ Hz (high frequency). We further investigated the interfacial impedance behavior using an equivalent circuit that contains a gel electrode interfacing with the epidermis and subcutaneous skin layers, where R$_d$ is the charge transfer resistance of the skin-electrode interface, C$_d$ is the double-layer capacitance between the skin and electrode. At the skin

level, R$_s$ and C$_s$ are the skin resistance and capacitance and R$_{tissue}$ is the deeper tissue resistance, respectively.

We examined the capability of VH20 hydrogel as a cathode in Mg battery (Fig. 3d). The open-circuit voltage for a compact Mg battery with polyvinyl alcohol (PVA)/PBS gel electrolyte was in the range of 1.48–1.46 V. It dropped immediately when a discharge current was applied and soon reached a flat discharge plateau. In this battery system, Mg was oxidized to Mg$^{2+}$ and reacted with hydroxide ions during the discharge, whereas the reduction of viologen at the cathode led to the expulsion of incorporated drugs. The overall reaction is as follows:

At anode:

$$Mg + 2OH^- \rightarrow Mg(OH)_2 + 2e^-$$

At cathode:

$$(VH)^{++}/2Dex^- + 2e^- \rightarrow (VH)^0 + 2Dex^-$$

Overall reaction:

$$Mg + (VH)^{++}/2Dex^- + 2H_2O \rightarrow Mg(OH)_2 + (VH)^0 + 2Dex^- + 2H^+$$

The mid-point discharge voltage ranged between 1.09 and 1.05 V, which was sufficient to drive the reduction of viologen (Supplementary Fig. 10a). The battery delivered a discharge capacity from 0.92 to 3.28 mAh cm$^{-2}$ at a current density of 10−100 μA cm$^{-2}$ (Fig. 3e). This battery exhibited slightly lower middle point voltage and capacity than the previously reported compact Mg battery[40] (1.29–1.06 V, 4.42−0.93 mA h cm$^{-2}$) with electrodeposited polypyrrole cathode due to the low redox voltage for viologen. It is also noted that this battery provided a relatively stable discharge plateau at all these current densities of 5−100 μA cm$^{-2}$ (Fig. 3f), indicative of good reversibility. EIS with an equivalent circuit model is used to gain a deeper insight into this electrochemical system. In the circuit, R$_s$ is the bulk resistance; R$_{ct}$ is the charge transfer resistance; constant phase element CPE is

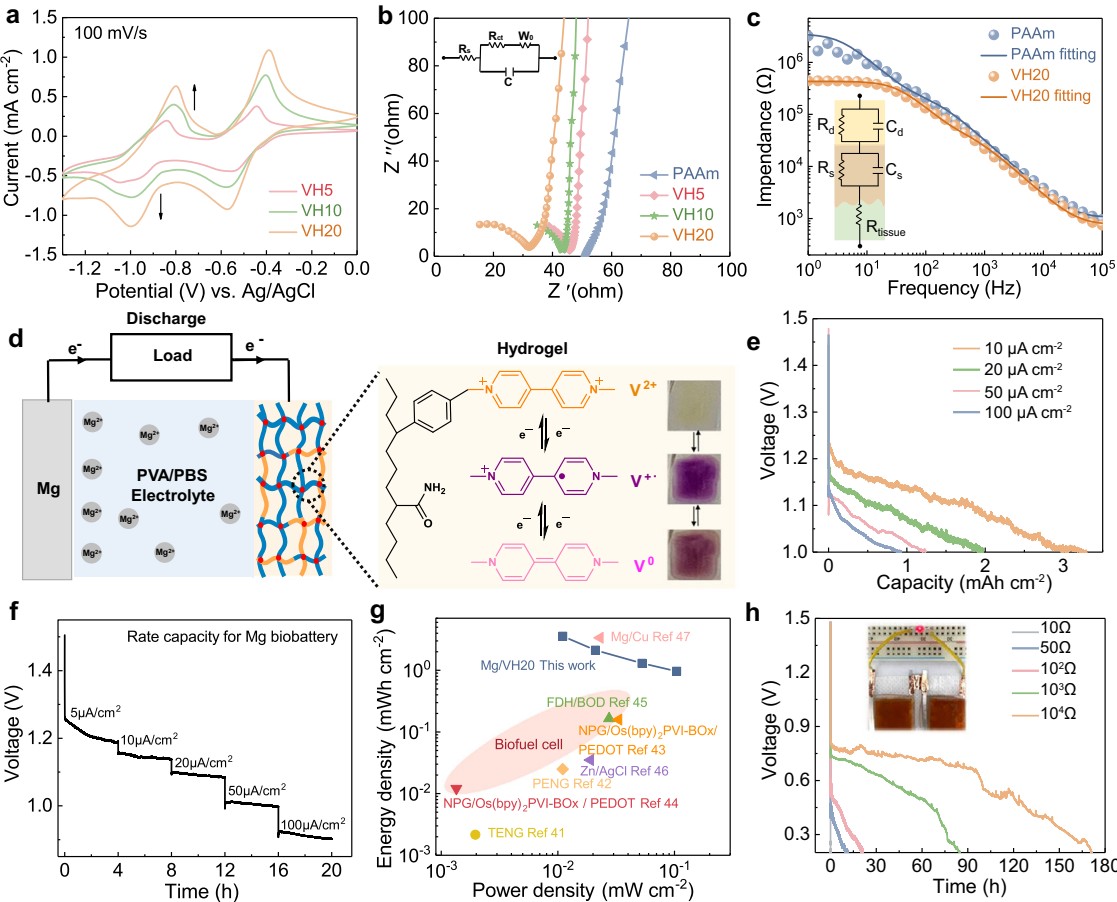

**Fig. 3 | Electrochemical properties of P(AM-co-SV) hydrogels and Mg batteries.**
**a** CV of P(AM-co-SV) hydrogels on ITO electrode in 0.01 M PBS solution with a scan rate of 100 mV s⁻¹. **b** Nyquist plots of P(AM-co-SV) hydrogels on ITO electrode in 0.01 M PBS solution. **c** EIS analysis of the tissue impedance of PAAm and VH20 hydrogels interfaced with the forearm skin (inset, equivalent circuit model). **d** Schematic diagram of the Mg battery using VH20 cathode and PVA/PBS gel electrolyte at various current densities. **e** Galvanostatic discharge curves and (**f**) rate capability of compact Mg batteries. **g** Ragone plot of compact Mg battery compared with the reported power sources for self-powered iontophoresis. **h** Time-resolved discharge curves of Mg batteries with various resistors, inset shows two batteries in series lighting up the diode.

associated with the double-layer capacitance across the electrode/electrolyte interface; $W_o$ corresponds to the Warburg impedance resulting from the semi-infinite diffusion of ions within the electrode at low frequency. The Nyquist plot demonstrated low bulk resistance of 121.4 Ω and charge transfer resistance of 60.4 Ω (Supplementary Fig. 10b), reflecting good electrical contact and low internal resistance.

The areal energy and power densities were used to build Ragone plots in comparison to other power sources for self-powered iontophoresis (Fig. 3g). The battery delivered an areal energy density of 3.57 mWh cm⁻² at an areal power density of 11 μW cm⁻², while an areal energy density of 0.97 mWh cm⁻² could still be delivered at a high areal power density of 104 μW cm⁻². The maximum energy density was higher than other power sources for iontophoresis, including triboelectric/piezoelectric nanogenerators[41,42], enzymatic biofuel cells[43–45], and galvanic cells[46,47]. The performance of the Mg battery in the iontophoresis patch was also evaluated under a load of resistors. It displayed a voltage of 0.62 V for up to 170 h with a resistor of 10⁴ Ω (Fig. 3h). Two Mg batteries connected in series could generate enough power to illuminate an LED bulb.

## On-demand drug release driven by an external source and iontophoresis

Dex was used as a model drug and counter-ion incorporating into the positively charged hydrogel to balance the charge[48]. Under the voltage stimulation of −1.0 V, pure PAAm hydrogel had almost no drug release

based on the absorbance at 242 nm (Supplementary Fig. 11). The drug release profiles of P(AM-co-SV) hydrogels at a controlled potential of −1.0 V (vs. Ag/AgCl) showed a high release rate in the initial state (40 min), accompanied by a subsequent plateau reaching an equilibrium in ca. 90 min (Fig. 4a). The electro-stimulated release efficiency of P(AM-co-SV) hydrogels was similar at −1.0 V, while the cumulative amount was highly dependent on the viologen contents (Fig. 4b). The electro-stimulated release profiles of VH20 hydrogel were investigated under natural diffusion, −0.6 V and −1.0 V, respectively (Fig. 4c). The spontaneous release of loosely bound and physically absorbed species was insignificant (6.98% after 180 min). The cumulative drug release was 2.08 mg g⁻¹ at −0.6 V (V⁺⁺ changed to radical V⁺), 3.08 mg g⁻¹ at −1.0 V (V⁺⁺ changed to V⁰) after 180 min. The appearance of VH20 hydrogel during the drug release could be used as a color-indicated smart drug release system (Supplementary Fig. 12)[49].

The properties and drug delivery applications of this redox-active polyelectrolyte hydrogel were compared with reported conductive hydrogels (Supplementary Table 3). Currently, most conductive hydrogel composites face the challenge of homogenously incorporating hydrophobic conductive components into hydrophilic hydrogels to form a percolating conductive network. Hence, conventional hydrogel drug reservoirs with low conductivity (below 1 mS cm⁻¹) and insufficient responsiveness to electrical stimuli require high potentials (above 1 V) to induce effective drug delivery[27,28,46,50,51]. Such high voltage may pose a potential health risk to the human body. On the other

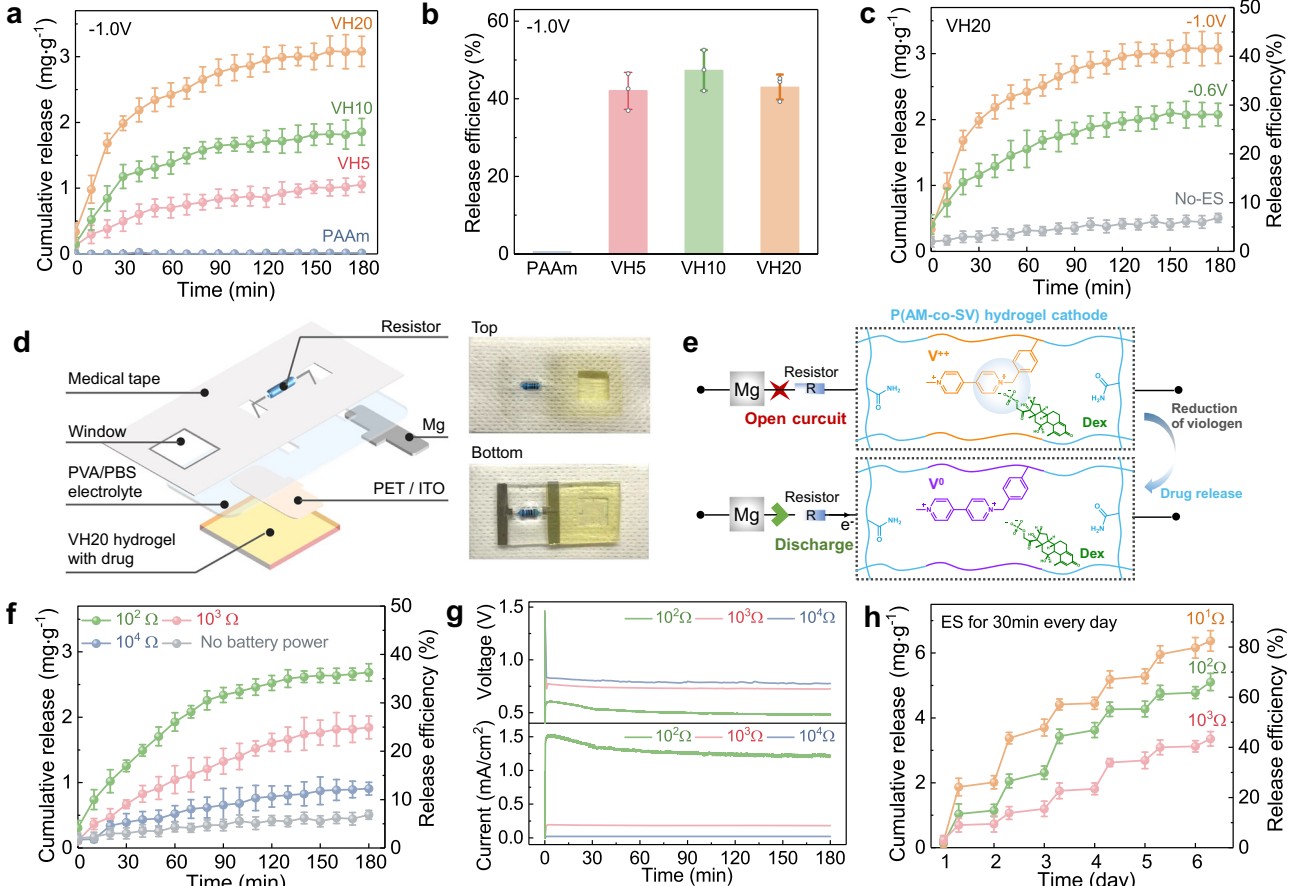

**Fig. 4 | On-demand drug release in PBS driven by an external source and iontophoresis. a** Cumulative amount, and (**b**) release efficiency of Dex from P(AM-co-SV) hydrogels at a controlled potential of −1.0 V for 180 min (*n* = 3 independent experiments). **c** Cumulative amount and release efficiency from VH20 hydrogel operating at diffusion (open-circuit mode), −0.6 V and −1.0 V for 180 min (*n* = 3 independent experiments). **d** Schematic circuit diagram and photographs of the wearable Mg battery-powered iontophoresis patch. **e** Drug release mechanism of the iontophoresis patch and corresponding molecular structure change of VH20 hydrogel at different stages. **f** Cumulative amount and release efficiency from Mg battery-powered iontophoresis patch with various external resistances (*n* = 3 independent experiments). **g** Electrochemical output profiles of Mg batteries during delivery with installed resistors. **h** Controlled intermittent release in 6 days from the battery-powered iontophoresis patch operating in "on-off" mode, with "on" representing discharge with various external resistances for 30 min every day, "off" representing the open-circuit mode (*n* = 3 independent experiments). Data are presented as mean ± SD in (**a**, **b**, **c**, **f**, **g**, and **h**).

hand, the covalently linking viologen into the PAAm backbone provided a prompt and stable response to electrical stimuli, enabling more precise control over the release kinetics. This polyelectrolyte hydrogel outperforms conductive hydrogel composites by leveraging a low driven potential (−0.6 and −1.0 V) for drug release, antibacterial activity to prevent infection, as well as its softness and low-impedance interface with roughed skin surfaces.

The Dex release was also investigated in a Mg battery-powered iontophoresis patch comprised of PVA/PBS gel electrolyte, VH20 hydrogel cathode, and an external resistor for generating controllable electric current. For maximal comfort and wearability, they were attached to an oxygen-permeable medical tape with two through-holes for installing resistors and observing the VH20 hydrogel (Fig. 4d). This Mg battery-powered release mechanism was associated with the redox processes of viologen. VH20 hydrogel was electrochemically reduced by Mg anode, with a transition from dication (V$^{++}$) to neutral form (V$^{0}$). When the battery was discharged, a concomitantly decreased electrostatic interaction and expulsion of Dex anions occurred, and the release rate can be regulated by the external resistors (Fig. 4e). A similar release trend was observed in the battery-powered patch compared with external sources. The release efficiency was approximately 25.8% when discharged with a 10$^3$ Ω resistor for 180 min (Fig. 4f), similar to 29% achieved at an applied potential of −0.6 V (Fig. 4c). The electrical profiles of the Mg battery with various installed resistors were also

recorded during the drug release. The battery output voltage dropped sharply, followed by an evident plateau, accompanied by a gradual increase in the cumulative drug release from the device (Fig. 4g). The maximum delivery rate of iontophoresis is restricted by the electric current tolerance, typically below 0.5 mA cm$^{-2}$, without causing skin irritation and pain. Specifically, the battery with a 10$^3$ Ω installed resistor generated a constant output voltage and current of 0.72 V and 0.18 mA cm$^{-2}$, respectively, which were sufficient and safe to transport the drug across the stratum corneum layer.

Since viologen could be re-oxidized by oxygen in the air during the resting process, the electrical profiles of the battery would recover to the initial levels (Supplementary Figs. 13 and 14), thus endowing a long-lasting release capability. The controlled intermittent release (with a tiny amount of passive delivery during the open-circuit state) was proven by an on-off operating sequence for the battery working alternately (30 min every day) (Fig. 4h). The battery with 10$^3$ Ω installed resistor obtained a cumulative drug release efficiency of 46.9% in 6 days, higher than that (25.8%) achieved for continuous delivery under the same electro-stimulation time (180 min). The detailed drug release profiles of VH20 hydrogel driven by an external source and battery-powered iontophoresis were tallied in Supplementary Table 4. Taken together, the delivery dosage can be precisely controlled by engineering the viologen hydrogel and iontophoresis stimulation.

## Transdermal delivery of the Mg battery-powered iontophoresis patch

Transdermal delivery of the iontophoresis patch involves two processes, on-demand drug release from the hydrogel and drug permeation across the skin. After studying the Mg battery-powered drug release, we further evaluated the cathodal transdermal delivery efficacy from the patch with the Dex-loaded VH20 hydrogel (without Mg battery power) as control (Fig. 5a)[19,20,44,52]. It was validated by applying the iontophoresis patch on hairless mouse skin using a Franz-type cell. Passive delivery of Dex-loaded VH20 hydrogel on the skin demonstrated a tiny permeation amount collected from the receptor chamber during 3 h operation (Fig. 5b). In contrast, cathodal iontophoresis resulted in significant skin permeation with a cumulative amount of 7.1–10.2 $\mu g\,cm^{-2}$ when discharged with installed resistors. It can be explained by the fact that the battery-empowered electrical current flew through the VH20 hydrogel, expelling Dex, which penetrated the skin by electrorepulsion force.

The intradermal delivery into porcine skin was carried out after a 3 h operation of the iontophoresis patch. We performed tape stripping and skin extraction studies to quantify the amount of Dex localized in the skin. Although a 3.2-fold increase in permeation amount (Fig. 5b) was observed for the patch (with a $10^3\,\Omega$ resistor) compared with the VH20 hydrogel (without Mg battery power), there was no significant difference in deposition amount in the stratum corneum (Fig. 5c). It may be explained by the fact that negatively charged Dex could not bind to enough sites in the stratum corneum due to electrorepulsion. In contrast, enhanced drug deposition was observed for the patch in the underlying skin compared to the passive delivery (without Mg battery power). Iontophoresis with a $10^3\,\Omega$ resistor significantly elevated the drug delivery into the deep tissue layers, as threefold increased amount of Dex was deposited within the epidermal/dermal layer compared with the VH20 hydrogel. Skin penetration was also visualized using sulforhodamine B (RB) as a model drug. RB mainly remained in the stratum corneum and epidermis after 30 min of passive delivery (without Mg battery power), while RB penetrated the stratum corneum into the dermal area from the patch (Fig. 5d). RB delivered by an iontophoresis patch (with a $10^3\,\Omega$ resistor) penetrated the skin with a thickness of ~2 times deeper than the VH20 hydrogel after 1 h. In conclusion, this Mg battery-powered iontophoresis patch is effective in delivering anionic drugs into and through the skin via the transdermal route.

## Therapeutic efficacy of wearable battery-powered iontophoresis patch

To further demonstrate the transdermal therapeutic efficacy, this wearable iontophoresis patch was applied to treat the imiquimod (IMQ)-induced psoriasis mouse model with on-demand delivery of Dex. Psoriasis is a typical chronic inflammatory skin disease that manifests as dry, thickened, erythematous, and scaly skin[53]. Dex has effectively treated allergic and autoimmune diseases due to its antibacterial, anti-inflammatory, and anti-allergic properties. After 6 days of modeling using IMQ, psoriasis mice exhibited dorsal skin folds, thickened stratum corneum, erythema and desquamation, and significant weight loss (Supplementary Fig. 15). Psoriasis symptoms disappeared from the iontophoresis patch-treated skin surface (Fig. 6a). As the electrochemical reaction proceeded after connecting to the resistor ($10^3\,\Omega$) for 180 s, a color change from yellow (off state, $V^{++}$ form) to purple (on state, $V^0$ form) was observed in the visual window of VH20 hydrogel (Fig. 6b). With this setup, a user can roughly gauge the patch working states without any external complex equipment, benefiting on-site healthcare fields.

Spleens and skin tissues were collected after the mice were treated and executed on the fifth day. Spleens treated with the iontophoresis patch demonstrated the same color and morphology as the control group and the smallest size compared with Dex solution and Dex-loaded VH20 hydrogel groups (Fig. 6c). Skin histopathology is the most intuitive evaluation method for psoriasis. The thickness of the lesioned skin returned to normal after device treatment (54 µm); however, the Dex group demonstrated a much thicker epidermal thickness of 77 µm with pronounced psoriasis symptoms (Fig. 6d). Comparing the hematoxylin and eosin (H&E)-stained skin tissue section photographs, the Dex group still exhibited prominent psoriasis features and inflammatory cell infiltration (yellow arrows in Fig. 6e). The thickened stratum corneum on the mice surface treated with VH20 hydrogel completely disappeared, and the stratum corneum returned to normal, but the slight epidermal hyperplasia (orange arrows in Fig. 6e) still existed. In contrast, the mice treated with the iontophoresis patch ultimately recovered to normal because Dex was delivered transdermally across the epidermal layer while epidermal moisture was maintained.

## Discussion

The iontophoresis patch offers great advantages of advancing compliance with patients' need for non-invasiveness and facilitating transdermal drug delivery without compromising efficacy. Current self-powered iontophoresis based on energy harvesting from biomechanical motion or biofuels is limited due to the low and unstable energy supply. Moreover, the majority of self-powered iontophoresis require complex electronic circuits with onboard batteries, which yield bulky and costly devices.

In this study, we have developed a wearable battery-powered iontophoresis patch with P(AM-co-SV) hydrogel as the integrated drug-loading cathode. This simplified configuration demonstrated three significant functions aiming toward efficient and controllable transdermal iontophoresis (comparison with previous self-powered iontophoresis in Supplementary Table 1): (1) Ion-conducting P(AM-co-SV) hydrogels were engineered for an optimal bioelectronic interface, including ultra-soft nature (elastic modulus of 0.4–1.2 kPa), low tissue-interface impedance ($4.3 \times 10^5\,\Omega$ at 1 Hz), good cytocompatibility and antibacterial properties. (2) Compact Mg battery with the hydrogel cathode as the built-in power source delivered a high energy density of $3.57\,mWh\,cm^{-2}$, higher than other reported self-powered iontophoresis, thus providing reliable and long-lasting energy output. (3) Battery-powered iontophoresis generated an on-demand transdermal release profile without complicated feedback circuits due to its straightforward potential-controlled release mechanism.

The patch-working states can also be readily indicated by the color change from the electrochromic hydrogel. Moreover, the therapeutic effects of psoriasis mouse therapy were evaluated in vivo. The dry, thickened, erythematous, and scaly skin reverted to normal after 5 days with the controllable release of Dex. This iontophoresis patch with the built-in battery may provide another way along the route to smart, personalized medical care.

## Methods

### Synthesis of P(AM-co-SV) hydrogels

Synthesis of 1-methyl-(4,4') bipyridinyl-1-ium iodide (VI): 4,4-Bipyridine (2.5 mmol, 0.39 g) and methyl iodide (1 mmol, 0.14 g) were mixed in dry dichloromethane, heated at 50 °C, and stirred for 48 h. The reaction mixture was cooled to room temperature, and the orange precipitate was collected by filtration. The yellow precipitate was washed with dichloromethane and ethyl acetate and then dried in a vacuum to give the first product VI. Yield: 78%. $^1H$ NMR (400 MHz, DMSO) δ 9.14 (d, $J = 6.8$ Hz, 2H), 8.87 (dd, $J = 4.5, 1.6$ Hz, 2H), 8.62 (d, $J = 6.8$ Hz, 2H), 8.04 (dd, $J = 4.5, 1.7$ Hz, 2H), 4.38 (s, 3H). $^{13}C$ NMR (101 MHz, DMSO) δ 152.31 (s), 151.48 (s), 146.62 (s), 141.30 (s), 125.42 (s), 122.34 (s), 48.08 (s). HRMS: m/z: calcd for $[M + H]^+$ $C_{19}H_{18}NO^+$: 171.0930, found: 171.0930.

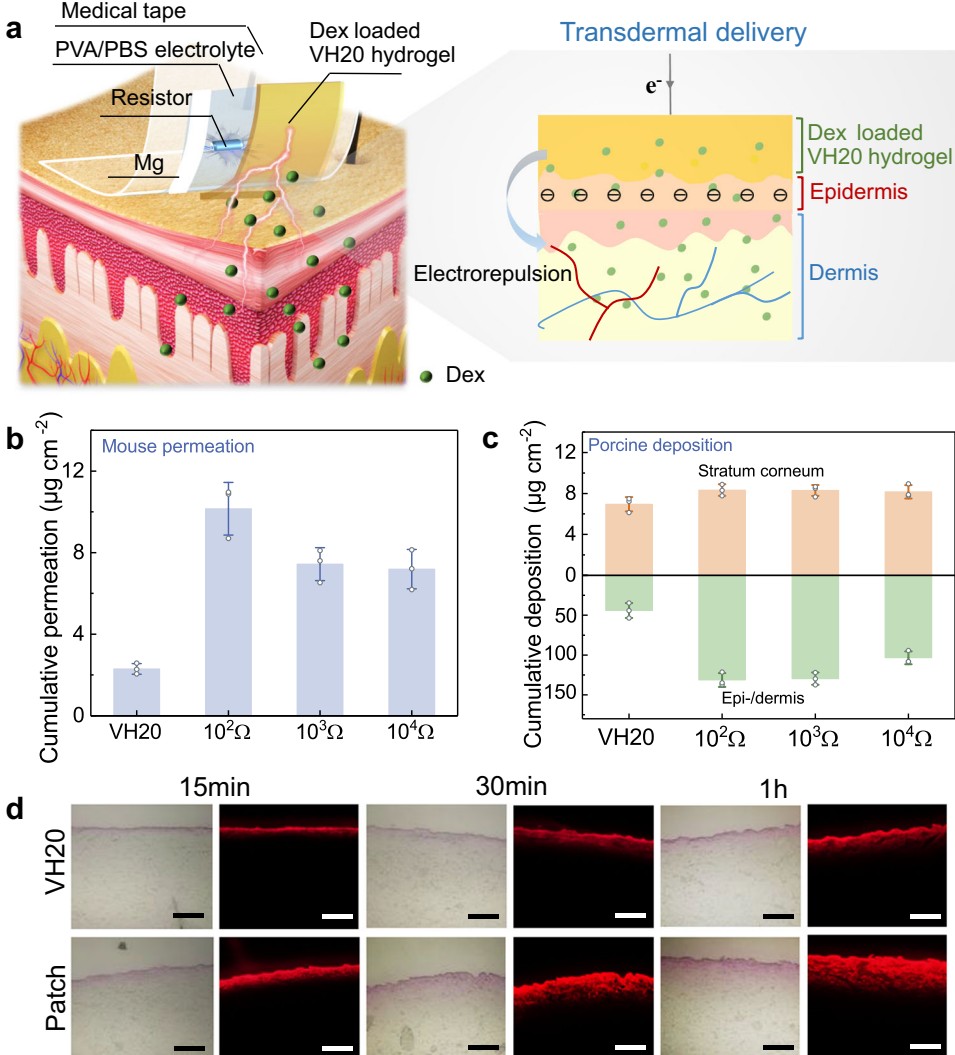

**Fig. 5 | Transdermal delivery of Dex from the Mg battery-powered iontophoresis patch. a** Scheme of a wearable iontophoresis patch with built-in Mg battery for transdermal delivery. **b** Cumulative permeation across the hairless mouse skin collected from the receptor chamber using a Franz diffusion cell after 3 h application. (*n* = 3 independent experiments). **c** Deposited amount within the stratum corneum and epi-/dermis after 3 h application to the porcine skin. (*n* = 3 independent experiments). **d** Optical and fluorescent cross-sectional images of porcine skin via passive delivery and iontophoresis patch (RB was used as a model drug for easy observation; scale bar: 200 μm. Measurements were repeated three times independently with similar results). Data are presented as mean ± SD in (**b**, **c**).

Synthesis of p-styrene-bipyridine (SV): VI (1 mmol, 0.298 g) and 4-chlorostyrene (1.5 mmol, 0.228 g) were mixed in dry dimethylformamide, heated at 65 °C and stirred for 48 h. The reaction mixture was cooled, and the dark orange precipitate was collected by filtration, recrystallized from ethyl acetate 3 times, and then dried in a vacuum to give the second product SV. Yield: 43%. $^1$H NMR (400 MHz, D$_2$O) δ 9.10 (d, *J* = 6.9 Hz, 2H), 8.99 (d, *J* = 6.7 Hz, 2H), 8.47 (dd, *J* = 13.1, 6.7 Hz, 4H), 7.56 (d, *J* = 8.2 Hz, 2H), 7.45 (d, *J* = 8.2 Hz, 2H), 6.77 (dd, *J* = 17.7, 11.0 Hz, 1H), 5.87 (t, *J* = 8.8 Hz, 3H), 5.34 (d, *J* = 11.0 Hz, 1H), 4.70 (s, 12H), 4.44 (s, 3H). $^{13}$C NMR (101 MHz, D$_2$O) δ 146.33 (s), 145.38 (s), 139.05 (s), 135.61 (s), 131.62 (s), 129.80 (s), 127.24 (s), 127.19 (d, *J* = 11.5 Hz), 126.69 (s), 116.01 (s), 64.41 (s), 48.44 (s). HRMS: m/z: calcd for [M + H]$^+$ C$_{19}$H$_{18}$NO$^+$: 276.1344, found: 276.1344.

Synthesis of P(AM-co-SV) hydrogels: SV, acrylamide, and N, N′-methylenebisacrylamide were dissolved in deionized water. KPS/TEMED was then used to initiate free radical polymerization. Hydrogels were formed after being left at room temperature for 2 h. For comparative experiments, hydrogels with different viologen contents (0%, 5%, 10%, 20% (wt%)) were prepared.

Loading of dexamethasone sodium phosphate: The freeze-dried hydrogel (1 g) was put in 100 mL Dex-containing PBS solution (100 μg mL$^{-1}$) and removed after 36 h. After removing the free drug, the drug-loaded hydrogels were obtained by immersing them in deionized water for 3 days. The drug-loaded hydrogels (0.02 g) were digested with inverse aqua regia for 7 days. The Dex loading capacity was measured from phosphorus elemental content in the hydrogel, obtained by an inductively coupled plasma-optical emission spectrometer (ICP, Agilent 725).

Characterizations: $^1$H NMR and $^{13}$C NMR of monomers were conducted with a Bruker 400 NMR spectrometer. The freshly prepared drug-loaded hydrogels were fully swollen in deionized water and then freeze-dried in liquid nitrogen. The internal morphology of freeze-dried hydrogel was obtained by a field emission SEM (FEI Nova NanoSEM 450). The infrared spectra of the freeze-dried hydrogel powder were collected by FTIR (Thermo Scientific Nicolet iS10). The emission spectra of hydrogels were collected by fluorospectro photometer (F97Pro, Lengguang).

Equilibrium swelling ratio: The swelling behavior of the freeze-dried hydrogel (1 g) was tested by soaking in a large amount of PBS

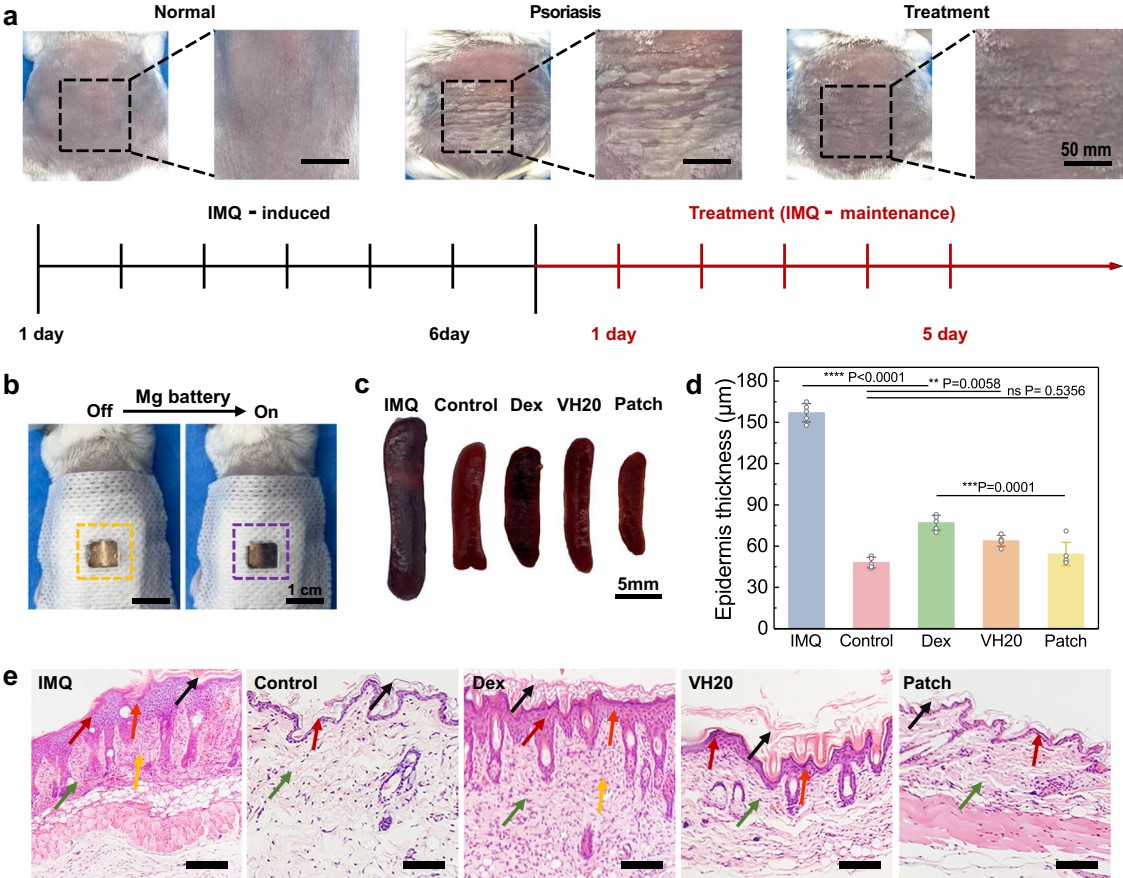

**Fig. 6 | Wearable Mg battery-powered therapeutic devices for psoriasis mouse treatment. a** Timeline and corresponding photos of normal skin, psoriasis model, and treatment process. **b** Color change of the Dex-loaded VH20 hydrogel during treatment; the dash area represents the patch size. **c** Mouse spleen photographs for IMQ-induced, and treated with Dex solution, Dex-loaded VH20 hydrogel, ionto-phoresis patch, normal model as the control. **d** Epidermis thickness of psoriasis mouse under different treatments. (*n* = 5 biologically independent animals). Data are presented as mean ± SD. *$P < 0.05$, **$P < 0.01$, ***$P < 0.001$, ****$P < 0.0001$, ns for no significance, $P$ value was generated by one-way analysis of variance (ANOVA), followed by Tukey's multiple-comparison post hoc test. **e** H&E staining of psoriasis mouse skin under different treatments, scale bar: 200 μm. Black arrows: hyper-keratosis; orange arrows: epidermal hyperplasia; red arrows: epidermis; green arrows: cortical layer; yellow arrows: inflammatory cell infiltration.

solution at room temperature until the swelling equilibrium was reached. The equilibrium swelling ratio (ESR) was calculated according to below equation:

$$ESR = \frac{(W_s - W_d)}{W_d} \quad (1)$$

where $W_s$ is the hydrogel weight after solubilized equilibrium, $W_d$ is the dry weight of the hydrogel.

Ionic conductivity: The luminescent diode photographs were obtained by varying the effective connection distance of the hydrogels (2, 1, and 0.5 cm) using two batteries (1.5 V) as the power source. The conductivity of hydrogels (1 × 1 × 0.3 cm) was measured through AC impedance (CHI660e) in the frequency range from 1000 kHz to 1 Hz. The electrical conductivity was calculated according to Eq. (2):

$$\sigma = \frac{1}{R}\frac{L}{A} \quad (2)$$

where $\sigma$ is the electrical conductivity (mS cm$^{-1}$), $R$ ($\Omega$) is determined from the intercept value of the AC impedance plot on the real axis (x-axis) at high frequency, $L$ and A represent the thickness and cross-sectional area of hydrogels, respectively.

## Mechanical properties

P(AM-co-SV) hydrogels were made into spindle-shaped specimens (30 × 10 × 2 mm). Mechanical properties were tested on a universal testing machine (Mark-10) equipped with a 100 N load cell. The tensile speed was fixed at 5 mm min$^{-1}$ to obtain the stress-strain curve. The elastic modulus was calculated from the slope of the uniaxial tensile curve in the strain range of 5–15%. Toughness was calculated from the integrated area of the stress-strain curve.

The energy dissipation behavior of VH20 hydrogel was investigated by examining the loading-unloading stretching curves at different strains and the continuous loading-unloading stretching curves at 100% strain. The integrated area of the closed region of the loading-unloading curve was defined as the energy dissipated by the hydrogel during the stretching process. The dynamic mechanical properties of hydrogels (8 mm diameter and 2.4 mm thickness) were tested by a rheometer (DHR-2, Waters Corporation) with a fixed strain of 0.1% and a frequency scan test range of 100–0.1 rad s$^{-1}$.

## Cytocompatibility and antibacterial properties

Cell culture with the hydrogel extracts: PAAm and VH20 hydrogels were soaked in 75% ethanol for 24 h, then washed with sterilized deionized water to remove the residual ethanol. The hydrogel extract was obtained by immersing the sterilized hydrogel (1 g) into a Dulbecco's modified Eagle's medium containing 10% fetal bovine serum

(10 mL) for 72 h. The concentrations of hydrogel extracts were adjusted to 0.02–0.1 g mL⁻¹. Mouse Fibroblasts Cells (L929, Cell Bioscience Inc) were seeded in a 96-well plate at a density of $5 \times 10^3$ cells per well with different hydrogel extracts. Cells were cultured in a humidified incubator at 37 °C with a 5% $CO_2$ atmosphere (MCO-18AIC). The cell viability in 96-well plates was examined by the MTT assay. At specific time intervals (1, 2, and 3 days) of cultivation, 10 µL of MTT was added to the culture medium, and the resulting solution was incubated at 37 °C for 4 h. The enzyme-labeled instrument was then used to measure the optical density at 490 nm (Bio-Tek Instruments).

Cell culture on the hydrogel surface: L929 cells were seeded at a density of $3 \times 10^4$ in a 24-well plate on sterilized hydrogels (PAAm and VH20). For fluorescence imaging, hydrogels were incubated in 2 µM Calcein AM and 4 µM propidium iodide in assay buffers for 30 min at 37 °C and in a 5% $CO_2$ atmosphere. The stained cells were observed directly under an inverted fluoresce microscope (DMIL, Leica). We further analyzed the cytocompatibility of the Mg battery by inoculating cells directly onto the VH20 hydrogel electrode when discharged with installed resistors for 30 min each day. The battery cytocompatibility was examined by cell staining and MTT assay after 3 days.

Antibacterial tests: Nutrient agar was used as a culture medium. Sterilized VH20 hydrogels (volumes of 1, 2, and 3 cm³) were co-cultured with *E. coli* and *S. aureus* in a liquid medium, respectively. The optical density values (OD600 nm) of the above co-cultures were measured by UV-Vis spectrophotometer after incubation in an automatic shaker at 37 °C for 24–96 h. The above co-cultured solution, after 24 h was diluted 10 times and inoculated onto a nutrient agar solid medium. The colony growth was examined after incubation in a 37 °C incubator for 24 h.

## Electrochemical properties of P(AM-co-SV) hydrogels and Mg battery

CV and EIS of hydrogels: The electrochemical properties of P(AM-co-SV) hydrogels were tested in a three-electrode system in 0.01 M PBS solution, with Pt wire as the counter electrode and Ag/AgCl as the reference electrode. The hydrogel working electrode was prepared by applying 40 µL of pre-polymerization solution on the ITO surface and then adding KPS to initiate the polymerization. A Gamry EIS 3000 system was operated to obtain EIS within the frequency range of 100 kHz to 0.01 Hz. To confirm the charge storage kinetics, the energy storage behavior can be revealed by below equation:

$$i = a \times v^{\wedge}b \qquad (3)$$

where $i$ is the peak current (mA cm⁻²) from the CV curve and $v$ is the scan rate (mV s⁻¹). To further fit the curve, the equation can be written as below equation:

$$Log(i) = b \times Log(v) + Log(a) \qquad (4)$$

The $b$ value is usually in the range of 0.5–1. When $b$ approaches 0.5, it is dominated by the diffusion-control process, which is close to a battery. When $b$ approaches 1, it is dominated by the surface-control process, which is close to a capacitor.

Tissue impedance: A two-electrode system was used for tissue impedance testing via an electrochemical workstation (CHI660E). VH20 hydrogel (8 mm diameter, 1 mm thickness) was used as the working electrode and counter electrode. Two circular electrodes were applied to the skin surface of the forearm in parallel (5 cm center distance).

Discharge of Mg battery: 9 g PVA was dissolved in 0.01 M PBS buffer solution (100 mL) at 90 °C for 3 h. The solution was cooled to room temperature before being transferred to the mold. PBS/PVA gel electrolytes were obtained after 3 freeze-thaw cycles, which were cooled down to -20 °C for 6 h, and raised to room temperature. The galvanostatic discharge and constant resistance discharge of the Mg battery with PBS/PVA gel electrolyte and VH20 hydrogel cathode were

performed on a Neware battery tester. They were placed into sealed bags to prevent water evaporation from the hydrogel. Galvanostatic discharge was applied until the cell voltage dropped to 1.0 V at current densities of 10–100 µA cm⁻². The energy and power density were determined from the mid-point of the galvanostatic discharge curves.

## On-demand drug release in PBS

Calibration curve: The absorbance-concentration calibration curve of Dex was established by recording the UV-Vis spectra of Dex-containing PBS solution between 210 and 290 nm. The released amount of Dex was determined by monitoring the absorption at 242 nm via UV-Vis spectroscopy.

External source-powered release: A piece of freeze-dried drug-loaded hydrogel (1.65 g) was sufficiently swollen before the release test. P(AM-co-SV) hydrogels loaded with Dex (1.8 mg cm⁻²) were immersed in the PBS solution. Electro-stimulated drug release was first performed with an external power supply through an electrochemical workstation in a three-electrode system. A potential of −1.0 or −0.6 V (*vs.* Ag/AgCl) was applied to the Dex-loaded hydrogel for 3 h. The PBS solution (2 mL) was removed at 10 min intervals, then the same volume of fresh PBS solution was added. The cumulative release amount into the PBS solution was calculated from the standard curve. The cumulative release efficiency is calculated according to below equation:

$$E_r = \frac{V_e \sum_{i}^{n-1} C_i + V_0 C_n}{m_{drug}} \times 100\% \qquad (5)$$

where $E_r$ is the cumulative release efficiency of the Dex, $V_e$ is the displacement volume of the PBS solution, $V_0$ is the total volume of release medium, $C_i$ is the release fluid concentration of the replaced PBS solution, $m_{drug}$ is the total drug-loading capacity, and $n$ is the number of PBS replacements.

Battery-powered release: The wearable iontophoresis patch was fabricated by attaching PBS/PVA gel electrolyte ($4 \times 2$ cm), Mg strip ($0.5 \times 2$ cm) anode, and VH20 hydrogel ($2.5 \times 2$ cm) cathode to the surface of commercial medical tapes (3 M). A piece of ITO/polyethylene terephthalate ($0.5 \times 2$ cm) was used as the current collector on VH20 hydrogel, which was connected to various resistors. The VH20 hydrogel cathode was in contact with the PBS solution. The iontophoresis patch with various installed resistors (10–10⁴ Ω) was applied continuously for 3 h. The cumulative drug release efficiency was determined from the Dex amount in PBS solution monitored by UV-Vis spectroscopy. The intermittent release from the iontophoresis patch (achieved by disconnecting the external resistors) was monitored daily for 6 consecutive days with 30 min electrical stimulation (discharge) every day.

## Transdermal delivery of the iontophoresis patch

Skin permeation test: A Franz diffusion cell was used for in vitro permeability experiments with hairless rat skin. Ten-week-old 300–350 g male hairless Spragur-Dawley rats (SD rats) were anesthetized with the inhalation of mixed isopentane gas (1–3%). Abdominal rat skin was freshly excised and cleaned thoroughly to remove the subcutaneous fat. The iontophoresis patch connected with resistors was placed over the skin tissue on the donor cell and discharged for 3 h. The permeation amount of Dex was collected from the receptor cell and analyzed using high performance liquid chromatography (HPLC, Shimadzu, UFLC, 20 A). The mobile phase (5 mM ammonium acetate and acetonitrile with a volume ratio of 1:1, 1 mL/min) was pumped through a $4.6 \times 250$ mm column packed with 5 µm C18 end-capped silica reversed-phase particles. The UV absorbance at 242 nm was used for detection.

Skin deposition test: To quantify the penetration of Dex into the skin, the iontophoresis patch was mounted on the fresh porcine skin and discharged for 3 h. Fresh porcine skin ($25 \times 25 \times 5$ mm) was washed and haircut, then immersed in PBS solution for 12 h at 4 °C. The

residual drug was subsequently washed away from the porcine skin's surface. The deposited drug was extracted from the stratum corneum by tape stripping (isolated stratum corneum from the dermis) and epidermal/dermal regions by skin extraction (chopped into small pieces and placed in fresh PBS solution). After centrifugation, the supernatant was collected, and the deposition amount was determined using HPLC. The deposition depth was visualized by monitoring sulforhodamine B (0.01 g within the VH20 hydrogel) into the porcine skin after 1 h application of iontophoresis with a passive delivery as control. The skin was then rinsed, frozen in liquid nitrogen, and sectioned into 15 µm thick specimens, followed by observing the cross-sectional area with optical and fluorescent microscopes.

## Animal tests

All in vivo experiments were approved by the Institutional Animal Care and Use Committee (IACUC) of Jilin University. The permit number is SY202206009. The feeding conditions are strictly following GB14925. All animal operations were performed under anesthesia, and every effort was made to minimize pain. Twenty-five BALB/c mice (5 weeks, gender random) were purchased from Liaoning Changsheng Biotechnology Co. Ltd. (Benxi, China) and randomly divided into 5 groups (each contained 5 mice). Mice were housed at $23 \pm 2$ °C, 50% humidity, 20 lux light intensity, and a light/dark alternation time of 12 h. IMQ (5%) cream was applied uniformly to the hair removal area ($2.5 \times 2.5$ cm) on the mice back at a dose of $65 \, mg \, cm^{-2}$ for 6 days. During treatment, IMQ was applied daily to maintain psoriasis. Except for the IMQ model and blank groups, the other three groups of mice were treated with 100 µL Dex solution ($0.5 \, mg \, mL^{-1}$), VH20 hydrogel (without electro-stimulation), and iontophoresis patch (with electro-stimulation for 30 min every day), respectively. After the mice were sacrificed, the spleen and skin of the treated area were excised and stored in the tissue fixative at 4 °C. Tissue skin was stained with H&E to observe the skin appearance and analyze the infiltration of inflammatory cells.

## Statistics and reproducibility

One-way analysis of variance (ANOVA) among multiple groups was performed for statistics. $P$ values were calculated by GraphPad Prism Software (version 9). The data in the figures were marked by * for $P < 0.05$, ** for $P < 0.005$, *** for $P < 0.001$, **** for $P < 0.0001$. All the data were reported as the mean ± SD. No animals were excluded from the analysis. No data were excluded from the analyses.

## Reporting summary

Further information on research design is available in the Nature Portfolio Reporting Summary linked to this article.

# Data availability

All the data in this study are available in the manuscript or Supplementary Information or source data. Correspondence and requests for materials should be addressed to Caiyun Wang. Source data are provided with this paper.

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

## Acknowledgements

This work is supported by the National Natural Science Foundation of China (521032089, 22275066, 21774046, 51602123), Jilin Provincial Science and Technology Department (20210508046RQ, 20200801057GH, 20190201049JC), Applied Basic Research Program of Changchun Municipal Science and Technology Project (21ZY22). C. Wang and G. Wallace acknowledge the support from the Australian Research Council Centre of Excellence Scheme (Project Number CE 140100012).

## Author contributions

X.J., D.C., C.W., and G.W. contributed to the idea and conceived the original project; Y.Z., X.J., S.J., and M.Z. and D.C. contributed to the experiment and analysis of data; D.P. and Y.T. performed the animal experiment; G.L. designed and fabricated the iontophoresis patch; Y.Z., X.J., C.W., and G.W. wrote and revised the manuscript.

## Competing interests

The authors declare no competing interests.
