## [Peer Review File · Nature Communications]

An integrated Mg battery-powered iontophoresis patch for efficient and controllable transdermal drug deliveryReviewers' Comments:

Reviewer #1:

Remarks to the Author:

This paper describes a Mg-Viologen battery that releases the counter anions from the polymeric Viologen cathode during the discharge of the battery. By using a drug (Dex) as the counter anion, the battery serves as a device for drug release to the skin surface. A viologen polymer was synthesized and its superior performance in releasing Dex was well studied. The cytocompatibility, antibacterial activity and larger release capacity of the synthesized viologen polymer would be of advantage in the application for skin patches. However, such the drug release from conducting polymer has been studied for long time; therefore this work does not contain enough novelty and new findings for publication in Nature Communications. This manuscript could be more suitable for a specialized journal after major revisions.

(1) Authors should stop using the term "biobattery" in the title and the text of the paper. Biobatteries are the devices that extract electrical energy using biochemical reactions of enzymes and microorganisms. On the other hand, the battery developed here is a Mg-Viologen primary galvanic cell.

(2) The electrophoretic penetration of anionic drugs from cathode is generally not efficient, since there are huge amount of smaller ions (Cl^- , Na^+ , K^+) that have higher mobility in skin. Therefore, the electroosmotic flow from anode to cathode becomes dominant for transport of larger molecules in skin. This is the conventional mechanism of reverse iontophoresis (collection of biomolecules in tissue at cathode). Author should revise the whole argument about the iontophoretic penetration from cathode. The "on-skin delivery" would be more suited for this report than the "transdermal delivery".

(3) In the present manuscript, the two factors (drug release and drug penetration) are mixed and making confusion. These should be studied separately with appropriate controls for each. Author should reconsider the experiments to demonstrate the effect of the developed patch. The effect of the electrically promoted penetration should be studied by the comparisons between the experiments using the same amount of drugs on skin with and without current. On the other hand, the effect of controlled release of drug will be discussed by comparing the batch coating of all drugs and the releasing of the same amount of drug to the surface of skin. However, the experiments in this manuscript are inappropriate for both. For example, in the Fig. 5g, only 50 μg Dex was coated, while the patch contains 1.8 mg (correct?) Dex.

(4) The bright field images are required to be added to Fig.5b. It's strange the boundaries of corneum / epidermis are not recognized in the fluorescent images. Also, it is known that the diffusion front of molecules of several hundred Da like rhodamine is hard to see due to their fast diffusion within skin.

Reviewer #2:

Remarks to the Author:

The authors proposed an iontophoresis patch with a built-in Mg biobattery based on a drug loaded P(AM-co-SV) hydrogel cathode, which could offer efficient transdermal drug delivery. The concept is novel and offers a new route to treat epidermal diseases. However, there are some important issues to be addressed:

(1) As the key point of the paper is the built-in bio-battery to promote transdermal drug delivery, some circuit models and schematic illustration are needed to clearly elucidate the working principle of this new iontophoresis patch. For example, the authors should label the signs of the electrodes and show why the drug will be driven toward the skin interface; what is the equivalent circuit at the drug loaded cathode/skin interface? Why is the extra resistor needed?

- (2) As the authors claim that the proposed iontophoresis patch can improved delivery efficiency (page 4 line 87), how does it compare to reported iontophoresis systems? Will part of the drug diffuse into the gel electrolyte and affect the efficiency?
- (3) Although PAAM is widely used, the residual monomers (acrylamide) that are not polymerized are considered toxic. The authors need to discuss its potential influence.
- (4) How did the author achieve the intermittent drug release? If it is achieved by disconnecting the resistor, this will cause inconvenience in practical application and irreversible connecting issues. In addition, disconnecting the resistor will not stop the biobattery, will it still drive drug diffusion to a certain extent?
- (5) How is the intermittent electrical profile (e.g., Fig. S13 C) measured? How to decide whether the recovery of the electrical curve comes from the disconnection or the reoxidization of viologen? Will the amount of intermittent drug release sufficient to treat associated disease?
- (6) Page 3 line 60, the authors states that "the commonly used hazardous materials (e.g., polyaniline, polypyrrole ...". However, polyaniline and polypyrrole are often considered to have low toxicity and are widely used as bio-interfaces. The authors should be careful about defining hazardous materials and sufficient literature evidence is needed.
- (7) Fig. 5g, is there any statistical difference between the VH20 and Patch groups regarding the epidermis thickness? If not, is the Patch group (with iontophoresis) advantageous compared to the Dex-loaded VH20 group (diffusion only)?
- (8) Page 13 Line 331 (Fig. 5b), the authors need to provide specific parameters for the iontophoresis, e.g., resistor and release time.
- (9) As the medical tape is gas permeable, will the hydrogel dry out in air?
- (10) Page 12, line 303, how is the drug release efficiency 26% calculated? The following statement cannot be obtained from Fig. 4g: "The release efficiency was approximately 26% when discharged with 1000 Ω installed resistors for 180 min, similar to 29% achieved for the potential controlled method at -0.6 V (Fig. 4g)". The authors should refer to Table S2 or Fig. 4C.
- (11) Page 10 line 263, why would the drug release profile reach a plateau?
- (12) Other minor issues: Fig. 1a, chemicals should be labeled with names; Fig. S5 a, the bar chart is not elucidative. Line chart is better; Page 10 line 255, and Fig S8 b, the type of LED that the biobattery can light up needs to be specified, as it indicates different level of output voltage.

RESPONSE TO THE REVIEWERS' COMMENTS

Reviewer #1:

This paper describe a Mg-Viologen battery that releases the counter anions from the polymeric Viologen cathode during the discharge of the battery. By using a drug (Dex) as the counter anion, the battery serves as a device for drug release to the skin surface. A viologen polymer was synthesized and its superior performance in releasing Dex was well studied. The cytocompatibility, antibacterial activity and larger release capacity of the synthesized viologen polymer would be of advantage in the application for skin patches. However, such the drug release from conducting polymer has been studied for long time; therefore this work does not contain enough novelty and new findings for publication in Nature Communications. This manuscript could be more suitable for a specialized journal after major revisions.

Response: Thank you for your critical comments. Although conducting polymers have been studied in electro-stimulated drug release, here we reported an intrinsic ion-conducting hydrogel by covalently linking redox active viologen to the PAAm backbone. This polyelectrolyte hydrogel outperforms conducting polymers by leveraging a low-driven potential for drug release, antibacterial activity to prevent infection, as well as its softness to form a conformal and low-impedance interface with skin surfaces.

As discussed in the Section *On-demand drug release driven by an external source and iontophoresis* on Page 10, conventional conducting polymer composite hydrogels are of low conductivity and insufficient responsiveness to electrical stimuli, which require high potentials to induce effective drug delivery (Supplementary Table S3). In contrast, this covalently linked polyelectrolyte hydrogel offers a low-driven potential for drug release and prompt stable response to electrical stimuli, enabling more precise control over the release kinetics.

Here we would like to re-emphasize the highlights of this work, which advances self-powered iontophoresis with efficient and controllable transdermal delivery.

1. As an electro-stimulated drug reservoir, this polyelectrolyte hydrogel offers an on-demand drug release profile and an optimal bioelectronic interface with ultra-soft nature and low tissue-interface impedance.
2. From the viewpoint of the new device structure, the integrated ion-conducting electrode decreases the system complexity and form factors, eliminating the interface impedance and associated Joule heating in the conventional design of a separate electrode and drug reservoir.
3. From the viewpoint of a new power source, the built-in Mg battery provides a reliable and efficient energy supply that enables high-dosage administration for long periods of operation.

To strengthen the novelty, we have re-edited the following text in the *Introduction* on Page 4 as follows:

Compared to the widely used conducting polymers for electro-stimulated drug release, this polyelectrolyte hydrogel outperforms conducting polymers by leveraging a low driven potential for drug release, antibacterial activity to prevent infection, as

well as an optimal bioelectrical interface for efficient transdermal delivery when deployed in the epidermal patch.

We have addressed all your specific comments below.

1. Authors should stop using the term “biobattery” in the title and the text of the paper. Biobatteries are the devices that extract electrical energy using biochemical reactions of enzymes and microorganisms. On the other hand, the battery developed here is a Mg-Viologen primary galvanic cell.

Response: Thanks for pointing it out. As suggested, we have changed the term *Mg biobattery* to *Mg battery*.

2. The electrophoretic penetration of anionic drugs from cathode is generally not efficient, since there are huge amount of smaller ions (Cl⁻, Na⁺, K⁺) that have higher mobility in skin. Therefore, the electroosmotic flow from anode to cathode becomes dominant for transport of larger molecules in skin. This is the conventional mechanism of reverse iontophoresis (collection of biomolecules in tissue at cathode). Author should revise the whole argument about the iontophoretic penetration from cathode. The “on-skin delivery” would be more suited for this report than the “transdermal delivery”.

Response: Thank you for your valuable suggestions. Dex is attached to the backbone of VH20 hydrogel via electrostatic interactions and released through the reduction of viologen. We believe that cathodal transdermal delivery is appropriate in this work from the following aspects.

1) Given that dexamethasone phosphate (Dex) is lipophilic and negatively charged, delivery from the cathode (with electrorepulsion as the main driving force) is more appropriate than from the anode (electroosmosis being the only available driving force). Previous studies have demonstrated the cathodal transdermal iontophoresis of Dex *in vitro* and *in vivo* (*Int. J. Pharm.* 2021, 607, 121009; *J. Control. Release* 2008, 131, 1).

2) To validate the cathodal transdermal delivery efficacy of the patch, we performed a skin permeation test using a Franz diffusion cell. *In vitro* skin permeation experiments revealed that the cathodal iontophoresis enhanced the cumulative amount of Dex permeating through the mouse skin compared to the passive delivery (without Mg battery power). The results have been presented in a new Fig. 5b, and the following text has been added on Page 12.

Transdermal delivery of the iontophoresis patch involves two processes, on-demand drug release from the hydrogel and drug permeation across the skin. After studying the Mg battery-powered drug release, we further evaluated the cathodal transdermal delivery efficacy from the patch with the Dex-loaded VH20 hydrogel (without Mg battery power) as control (Fig. 5a). It was validated by applying the iontophoresis patch on hairless mouse skin using a Franz-type cell. Passive delivery of Dex-loaded VH20 hydrogel on the skin demonstrated a tiny permeation amount

collected from the receptor chamber during 3 h operation (Fig. 5b). In contrast, cathodal iontophoresis resulted in significant skin permeation with a cumulative amount of 7.17-10.15 $\mu\text{g cm}^{-2}$ when discharged with installed resistors. It can be explained by the fact that the battery-empowered electrical current flew through the VH20 hydrogel, expelling Dex, which penetrated the skin by electrorepulsion force.

Fig. 5 Transdermal delivery of Dex from the Mg battery-powered iontophoresis patch. **(a)** Scheme of a wearable iontophoresis patch with built-in Mg battery for transdermal delivery. **(b)** Cumulative permeation across the hairless mouse skin collected from the receptor chamber using a Franz diffusion cell after 3 h application. **(c)** Deposited amount within the stratum corneum and epi-/dermis after 3 h application to the porcine skin.

3) To evaluate the influence of Cl^- , drug-loaded hydrogels were immersed in NaCl solutions (0.45%, 0.9%, and 1.8%). The cathodal delivery of Dex was decreased with 1.8% of NaCl (Fig. R1), but increased instead in low concentrations (0.45% and 0.9%). This may be caused by the enhanced ionic conductivity, which improves the ionic current and promotes drug release.

Fig. R1 Cumulative permeation across the hair-less mouse skin from the iontophoresis patch ($10^3 \Omega$) with different concentrations of NaCl-containing VH20 hydrogel.

3. In the present manuscript, the two factors (drug release and drug penetration) are mixed and making confusion. These should be studied separately with appropriate controls for each. Author should reconsider the experiments to demonstrate the effect of the developed patch.

1) The effect of the electrically promoted penetration should be studied by the comparisons between the experiments using the same amount of drugs on skin with and without current.

2) On the other hand, the effect of controlled release of drug will be discussed by comparing the batch coating of all drugs and the releasing of the same amount of drug to the surface of skin.

3) However, the experiments in this manuscript are inappropriate for both. For example, in the Fig. 5g, only 50µg Dex was coated, while the patch contains 1.8 mg (correct?) Dex.

Response: Thank you for your valuable suggestions. We have added more experiments, and the results are shown in a new Figure 5. A new Section *Transdermal delivery of the Mg battery-powered iontophoresis patch* has been added on Page 12 along with this new figure.

1) To demonstrate the drug penetration, we have supplemented the tape stripping and skin extraction tests for no-drug loaded hydrogel (to exclude the drug release process) with and without Mg battery powered. A 100 µL of Dex solution (30 mg/mL) was applied on the porcine skin, similar to the release amount (1.5-4.5 mg with external resistors, Fig 4f) from the iontophoresis patch. Enhanced local deposition in underlying skin (penetration into the epi-/dermis) was shown for the patch (with current) compared to the passive delivery (without current) (Fig. R2).

Fig. R2 Cumulative deposition into the porcine skin from the iontophoresis patch (no-drug loaded hydrogel) with the 3 mg of drugs on the skin.

We have also conducted the drug extraction from the porcine skin using the drug-loaded hydrogel. A new Fig. 5c has been added, and the following text has been revised on Page 12, in the section “*Transdermal delivery of the Mg battery-powered iontophoresis patch*”.

The intradermal delivery into porcine skin was carried out after a 3 h operation of the iontophoresis patch. We performed tape stripping and skin extraction studies to quantify the amount of Dex localized in the skin. Although a 3.2-fold increase in permeation amount (Fig. 5b) was observed for the patch (with a 10³ Ω resistor) compared with the VH20 hydrogel (without Mg battery power), there was no significant difference in deposition amount in the stratum corneum (Fig. 5c). It may be explained by the fact that negatively charged Dex could not bind to enough sites in the stratum corneum due to electrorepulsion. In contrast, enhanced drug deposition was observed for the patch in the underlying skin compared to the passive delivery (without Mg battery power). Iontophoresis with a 10³ Ω resistor significantly elevated the drug delivery into the deep tissue layers, as 3-fold increased amount of Dex was deposited within the epidermal/dermal layer compared with the VH20 hydrogel.

Fig. 5 (c) Deposited amount of Dex within the stratum corneum and epi-/dermis after 3 h application to the porcine skin

2) We have measured and compared the on-demand Dex release to PBS solution from Dex-loaded hydrogels (powered by an external power source) and the patch (powered by the Mg battery) in Fig. 4. Delivery of other anion drugs was not investigated in this study, but we are carrying out delivering adenosine triphosphate disodium salt using the iontophoresis patch with a new viologen hydrogel in our next work. The results demonstrate similar release behaviors.

3) Fig. 5g (Fig. 6d in the revised manuscript) did not show the drug penetration data but described the epidermis thickness of the psoriasis mouse under different treatments. It was 50 μm thick after the iontophoresis patch.

4. The bright field images are required to be added to Fig.5b. It's strange the boundaries of corneum / epidermis are not recognized in the fluorescent images. Also, it is known that the diffusion front of molecules of several hundred Da like is hard to see due to their fast diffusion within skin.

Response: Thank you for your valuable suggestions. To visualize the penetration of rhodamine B into the porcine skin, the cross-sectional optical and fluorescent images were captured at different discharge times. The results are shown in the new Fig. 5d and the following text has been added on Page 13.

Skin penetration was also visualized using sulforhodamine B (RB) as a model drug. RB mainly remained in the stratum corneum and epidermis after 30 min of passive delivery (without Mg battery power), while RB penetrated the stratum corneum into the dermal area from the patch (Fig. 5d). RB delivered by an iontophoresis patch (with a $10^3 \Omega$ resistor) penetrated the skin with a thickness of ~ 2 times deeper than the VH20 hydrogel after 1 h. In conclusion, this Mg battery-powered iontophoresis patch is effective in delivering anionic drugs into and through the skin via the transdermal route.

Fig. 5 (d) Optical and fluorescent cross-sectional images of porcine skin via passive delivery and iontophoresis patch (RB was used as a model drug for easy observation; scale bar: 200 μm).

The following text has been added to the *Experimental* Section on Page 21.

Transdermal delivery of the iontophoresis patch

Skin permeation test: A Franz diffusion cell was used for *in vitro* permeability experiments with hairless rat skin. Ten-week-old 300-350 g male hairless rats were anesthetized with the inhalation of mixed isopentane gas (1-3%). Abdominal rat skin was freshly excised and cleaned thoroughly to remove the subcutaneous fat. The iontophoresis patch connected with resistors was placed over the skin tissue on the donor cell and discharged for 3 h. The permeation amount of Dex was collected from the receptor cell and analyzed using high performance liquid chromatography (HPLC, Shimadzu, UFLC, 20A). The mobile phase (5 mM ammonium acetate and acetonitrile with a volume ratio of 1:1, 1 mL/min) was pumped through a 4.6×250 mm column packed with 5 μ m C18 end-capped silica reversed-phase particles. The UV absorbance at 242 nm was used for detection.

Skin deposition test: To quantify the penetration of Dex into the skin, the iontophoresis patch was mounted on the fresh porcine skin and discharged for 3 h. Fresh porcine skin ($25 \times 25 \times 5$ mm) was washed and haircut, then immersed in PBS solution for 12 h at 4 °C. The residual drug was subsequently washed away from the porcine skin's surface. The deposited drug was extracted from the stratum corneum by tape stripping (isolated stratum corneum from the dermis) and epidermal/dermal regions by skin extraction (chopped into small pieces and placed in fresh PBS solution). After centrifugation, the supernatant was collected, and the deposition amount was determined using HPLC. The deposition depth was visualized by monitoring sulforhodamine B (0.01 g within the VH20 hydrogel) into the porcine skin after 1 h application of iontophoresis with a passive diffusion as control. The skin was then rinsed, frozen in liquid nitrogen, and sectioned into 15 μ m thick specimens, followed by observing the cross-sectional area with optical and fluorescent microscopes.

Reviewer #2:

The authors proposed an iontophoresis patch with a built-in Mg biobattery based on a drug loaded P(AM-co-SV) hydrogel cathode, which could offer efficient transdermal drug delivery. The concept is novel and offers a new route to treat epidermal diseases. However, there are some important issues to be addressed:

Response: Thank you for your recognition of this work. We have addressed all the comments and added the suggested characterizations.

1. As the key point of the paper is the built-in bio-battery to promote transdermal drug delivery, **1)** some circuit models and schematic illustration are needed to clearly elucidate the working principle of this new iontophoresis patch. For example, the authors should label the signs of the electrodes and show why the drug will be driven toward the skin interface; **2)** what is the equivalent circuit at the drug loaded cathode/skin interface? **3)** Why is the extra resistor needed?

Response: Thank you for the valuable suggestions. We have revised the figures as suggested.

1) New figures illustrating the circuit model and drug release mechanism have been added as Fig. 4d and 4e. The following text has been added on Page 11.

For maximal comfort and wearability, they were attached to an oxygen-permeable medical tape with two through-holes for installing resistors and observing the VH20 hydrogel (Fig. 4d). This Mg battery-powered release mechanism was associated with the redox processes of viologen. VH20 hydrogel was electrochemically reduced by Mg anode, with a transition from dication (V^{++}) to neutral form (V^0). When the battery was discharged, a concomitantly decreased electrostatic interaction and expulsion of Dex anions occurred, and the release rate can be regulated by the external resistors (Fig. 4e).

Fig. 4 (d) Schematic circuit diagram and photographs of the wearable Mg battery-powered iontophoresis patch. **(e)** Drug release mechanism of the iontophoresis patch and corresponding molecular structure change of VH20 hydrogel at different stages.

2) The equivalent circuit at the drug-loaded cathode/skin interface has been added as an inset in Fig. 3c. The following text has been added on Page 8.

We further investigated the interfacial impedance behavior using an equivalent circuit that contains a gel electrode interfacing with the epidermis and subcutaneous skin layers, where R_d is the charge transfer resistance of the skin-electrode interface, C_d is the double-layer capacitance between the skin and electrode. At the skin level, R_s and C_s are the skin resistance and capacitance and R_{tissue} is the deeper tissue resistance, respectively.

Fig. 3c EIS analysis of the tissue impedance of PAAm and VH20 hydrogels interfaced with the forearm skin (inset, equivalent circuit model).

3) The use of external resistors enables the Mg battery function by completing the circuit for generating electric currents and tuning the release profile. The following text has been revised on Page 11.

The Dex release was also investigated in a Mg battery-powered iontophoresis patch comprised of PVA/PBS gel electrolyte, VH20 hydrogel cathode, and an external resistor for generating controllable electric current.

2. As the authors claim that the proposed iontophoresis patch can improved delivery efficiency (page 4 line 87), 1) how does it compare to reported iontophoresis systems? 2) Will part of the drug diffuse into the gel electrolyte and affect the efficiency?

Response: Thank you for your comments.

1) We have revised Table S1 “*Summary of current iontophoresis systems*” to compare with the reported electro-stimulated drug delivery. This iontophoresis patch demonstrated a stable energy output and an enhanced release capacity compared with energy harvester strategies. The continuous release efficiency for 3 h was in the range of 111.5-139.6 $\mu\text{g cm}^{-2}$, comparable to that using an external power supply (DC method, Fig. 4). The following text has been revised to *Introduction* on Page 4.

This wearable Mg battery-powered iontophoresis demonstrates stable current outputs and enhanced release capacity compared with energy harvester strategies (Supplementary Table S1), providing new therapeutic approaches for chronic skin diseases requiring precise drug delivery.

Supplementary Table S1 Summary of current iontophoresis systems.

Category	Typical materials	Output	Model drug	Release capacity	Application	Ref.
Direct current (DC)	—	0.3 mA	Dex	71 nmol h ⁻¹	Study the competition of chloride released from Ag/AgCl cathode on the iontophoretic delivery.	1
	Microneedle array	0.5, 1.0 mA	Insulin nanovesicle	—	A transdermal patch can achieve synergistic and remarkable enhancement of drug delivery with precise electronic control.	2
	Poly(ethylene glycol) hydrogel	100 mA cm ⁻²	Dextran/ Bevacizumab	1.443/0.9 mg cm ⁻²	An iontophoresis device demonstrates high-efficiency intraocular delivery.	3
	PPy nanoparticles	0.13 mA cm ⁻²	Insulin	68.29 μg cm ⁻²	Investigate PPy nanoparticles for controlled transdermal iontophoresis of insulin.	4
PENG	Poly(lactic acid-gold-PPy) microneedles	100 V, 2 μA	Dex	96.75 ng per needle	Collect and convert biomechanical energy into electrical energy to control drug release for psoriasis treatment.	5
TENG	DOX loaded red blood cells on the Cu film	70 V, 0.5 μA	DOX	59.7%	Killing cancer cells in vitro and in vivo at a low drug dosage.	6
	Poly(3-hexylthiophene) films	647 V, 165 μA	Salicylic acid	0.86 μg mL ⁻¹	TENG can provide a steady voltage supply for sustainable drug release.	7
	PPy/Dex film on gold electrode	100 V, 19 mW cm ⁻²	Dex	35 μg cm ²	Electricity generated from TENG was used to power iontophoresis treatment.	8
	Silicon nanoneedles-array	20 V, 4 μA	siRNA/dextran-FITC	82%/86%	TENG-driven electroporation system is developed for intracellular drug delivery in vivo and in vitro.	9
	Poloxamer hydrogel	1200 V, 20 μA	Rhodamine 6G/methylene blue	5.5 nmol/14.2 nmol	A wearable TENG is used as the motion sensor and energy harvester for iontophoresis.	10
	Polydimethylsiloxane drug reservoir	15 V, 1.5 mA	Fluorescent particles	5.3-40 μL min ⁻¹	TENG-based self-powered implantable drug-delivery system demonstrates its functionality for ocular drug delivery.	11
Biofuel cell	Biocathode: PEDOT functionalized gold electrode Bioanode: Carbon nanotubes	0.4 V, 33 mW cm ⁻²	Acetaminophen	—	A biocomputing, logic-based detection method with a controlled-release drug delivery actuator.	12
	Biocathode: PEDOT doped NPG/Os(bpy) ₂ PVI-BOx Bioanode: NPG/Os-(bpy) ₂ PVI-GOx	0.377 V, 1.35 μW cm ⁻²	Ibuprofen/Fluorescein/diamidinophenylindole	570 μg cm ⁻² /101 ng cm ⁻² /102 ng cm ⁻²	A self-powered, controlled drug-release system based on bilayer-modified electrodes has been demonstrated.	13
	Biocathode: BOD-carbon fabrics Bioanode: FDH-carbon fabrics	0.55-0.7 V, 10-50 μA cm ⁻²	Rhodamine B Ascorbyl glucoside	—	BFC-driven current-assisted penetration of ascorbyl glucoside and rhodamine B into the skin.	14
	Biocathode: PSp/carbon sphere/glassy carbon Bioanode: PDS/gold nanobowl/glassy carbon	145 μW cm ⁻²	DOX	62%	A robust glucose/O ₂ fuel cell-based biosensor is integrated with a targeted drug delivery system to create a self-sustained and highly compact drug delivery model.	15
	Biocathode: BOD-immobilized electrode Bioanode: FADGDH immobilized electrode	0.40 V, 157 μW cm ⁻²	—	—	An autonomous, self-powered, sensing actuator that employs the principle of bio-capacitor as the core technology.	16
Battery	Cathode: PEDOT Anode: Zn	—	Rhodamine B	—	A battery-driven drug delivery device powered by physiological	17

					pH can target intended sites and be actuated galvanically to trigger localized drug release.	
	Cathode: PPy Anode: Mg	—	Adenosine triphosphate (ATP)	92%	The drug release from this system can be performed without the external power source, and the massive drug release only occurs at around human body temperature.	18
	Cathode: AgCl Anode: Zn Drug-loading material: PEDOT /PAAm hydrogel	1.08 V, 0.2 mA, 60 μ W	Rhodamine B	—	Iontophoretic drug delivery device, and an electrical nerve stimulator, for transcutaneous bioanalysis and therapy.	19
	Cathode: PEDOT Anode: Zn	—	Gold nanoparticle	—	In vivo studies of PEDOT/Zn artificial micromotors demonstrate their distribution, retention, and toxicity.	20
	Cathode: P(AM-co-SV) hydrogel Anode: Mg	1.1 V, 10 mA cm^{-2} , 3.57 mWh cm^{-2}	Dex	139.6 $\mu\text{g cm}^{-2}$	Battery-powered iontophoresis generated an on-demand transdermal release profile without complicated feedback circuits.	This work

Notes

DC: direct current; PENG: piezoelectric nanogenerator; TENG: triboelectric nanogenerator; BFC: enzymatic biofuel cell;

NPG/Os(bpy)₂PVI-BOx: Os(bpy)₂PVI-mediated bilirubin oxidase (BOx); NPG/Os-(bpy)₂PVI-GOx: [Os(2,2'-bipyridine)₂(polyvinylimidazole)₁₀Cl]⁺²⁺ (Os(bpy)₂PVI)-mediated glucose oxidase (GOx);

BOD: bilirubin oxidase; FDH: fructose dehydrogenase; PSp: phosphatidylserinebinding peptide; PDS: partial complementary DNA double strand-decorated; FADGDH: flavin adenine dinucleotide-dependent glucose dehydrogenase;

PPy: polypyrrole; PEDOT: poly(3,4-ethylenedioxythiophene); PANI: polyaniline; PAAm: polyacrylamide;

Dex: Dexamethasone sodium phosphate; DOX: Doxorubicin

2) Yes, a small amount of Dex will diffuse into the electrolyte. When the patch was loaded with a 1000 Ω resistor for 3 h, the drug amount diffused into the PVA-PBS gel electrolyte was 0.23 mg g^{-1} (Fig. R3), which was 3.2% of the total drug release amount (2.68 mg g^{-1} , Fig. 4f) into the PBS solution. Therefore, the drug content in the gel electrolyte has a negligible effect on the release efficiency.

Fig. R3 Cumulative release into the gel electrolyte from the iontophoresis patch with different external resistors after 3 h of operation.

3. Although PAAM is widely used, the residual monomers (acrylamide) that are not polymerized are considered toxic. The authors need to discuss its potential influence.

Response: Thanks for the comments. As stated in the *Experimental* Section, the hydrogel was immersed in PBS to extract the residual monomers before use. We have tested the cell viability of VH20 hydrogel extracts (Fig. 2a). Here we further evaluated the toxicity of PAAm hydrogels and presented a new Fig. S5b in Supplementary information. The hydrogel extracts demonstrated a cell viability of 96%, similar to the cell cultured in the DMEM solution. The results revealed that the treated hydrogels showed no cytotoxic effect.

Supplementary Fig. S5b Cell viability of L929 cells cultured with PAAm hydrogel extracts (0.1 g/mL) for 1, 2, and 3 days.

4. 1) How did the author achieve the intermittent drug release? If it is achieved by disconnecting the resistor, this will cause inconvenience in practical application and irreversible connecting issues. 2) In addition, disconnecting the resistor will not stop the biobattery, will it still drive drug diffusion to a certain extent?

Response: 1) The intermittent drug delivery was achieved by disconnecting the resistor. Intermittent drug delivery may be further designed by introducing a switch in the circuit for practical applications as reported (*Adv. Funct. Mater.* 2020, 30, 1909886; *Nano Energy* 2019, 62,610-619). The following text has been revised to *Experimental* Section on Page 21.

The intermittent release from the iontophoresis patch (*achieved by disconnecting the external resistors*) was monitored daily for 6 consecutive days with 30 min electrical stimulation (discharge) every day.

2) Disconnecting the resistor will stop the Mg battery function while the passive diffusion occurs, releasing a much smaller amount of Dex than the battery-powered delivery. The cumulative passive release amount was in the range of 0.05-0.16 mg g⁻¹ each day (Fig. 4h), significantly smaller than the Mg battery-powered intermittent release (0.52-0.93 mg g⁻¹ during 30 min discharge with different external resistors). The following text has been revised on Page 11.

The controlled intermittent release (*with a tiny amount of passive delivery during the open-circuit state*) was proven by an on-off operating sequence for the battery working alternately (30 min every day) (Fig. 4h).

5. 1) How is the intermittent electrical profile (e.g., Fig. S13 c) measured? 2) How to decide whether the recovery of the electrical curve comes from the disconnection or

the reoxidization of viologen? **3) Will the amount of intermittent drug release sufficient to treat associated disease?**

Response:

1) The intermittent electrical profile in Fig. S13c was recorded directly from a Neware battery tester.

2) To investigate the electrical recovery, we tested the UV-Vis spectra of VH20 hydrogel at different discharge states (Supplementary Fig. S13). When the battery was discharged, the hydrogel exhibited a high absorption peak at 535 nm associated with the reduction of viologen (V⁰). After 6 min of power-off (recovery stage), the absorption peak at 535 nm disappeared, and the absorption curve returned to the initial state, indicating the re-oxidization of viologen (V²⁺). These results demonstrate that the recovery of the electrical curve is caused by the re-oxidization of viologen.

Supplementary Fig. S13 Absorption spectra of the VH20 electrode at different recovery times after the Mg battery was discharged for 30 min.

3) The intermittent drug release is sufficient to treat psoriasis disease.

The Dex content in the commercially available cream (Dexamethasone acetate cream, Sanyi) was 1 mg g⁻¹. The daily therapeutic dose of intermittent release from the patch can reach 1.06 mg g⁻¹. In addition, the drug loading content in the patch was 7.16 mg g⁻¹, which can provide enough release capacity. Therefore, intermittent drug release was sufficient for the disease treatment.

In addition, the therapeutic effect of the patch was confirmed in animal experiments in this work (Fig. 6), where the psoriasis mouse model showed normalization of epidermal thickness, the disappearance of scales, and reduction of inflammatory cells after 5 days of treatment.

6. Page 3 line 60, the authors states that “the commonly used hazardous materials (e.g., polyaniline, polypyrrole ...” However, polyaniline and polypyrrole are often considered to have low toxicity and are widely used as bio-interfaces. The authors should be careful about defining hazardous materials and sufficient literature evidence is needed.

Response: Thanks for pointing out our mistake. Conducting polymers are biocompatible biomaterials and widely used in the electro-stimulated drug release and

tissue engineering field. We have deleted the negative statement of conducting polymers as hazardous materials.

7. Fig. 5g, is there any statistical difference between the VH20 and Patch groups regarding the epidermis thickness? If not, is the Patch group (with iontophoresis) advantageous compared to the Dex-loaded VH20 hydrogel (diffusion only)?

Response: Thanks for the comments. There was no statistical difference between the VH20 (64 μm) and patch groups (49 μm) in terms of epidermal thickness (original Fig. 5g, new Fig. 6d). This similar epidermal thickness may be caused by the relatively high water content of the hydrogel. This moist environment contributes to the disappearance of psoriasis surface scales.

However, the H&E staining of the skin tissue graph (original Fig. 5h, new Fig. 6e) demonstrated that the epidermal scales in the patch group disappeared and the inflammatory cell number inside the tissue reduced. Therefore, the treatment effect of the patch group is advantageous compared to the VH20 hydrogel.

8. Page 13 Line 331 (Fig. 5b), the authors need to provide specific parameters for the iontophoresis, e.g., resistor and release time.

Response: Thanks for the valuable suggestions. A resistor of 1000 Ω and a release time of 1 h were used to visualize the penetration of rhodamine B into the porcine skin. As suggested by another reviewer, we have recaptured the cross-sectional optical and fluorescent images at different discharge times. The results are shown in the new Fig. 5d, and the following text has been added on Page 13.

Skin penetration was also visualized by using sulforhodamine B (RB) as a model drug. RB mainly remained in the stratum corneum and epidermis after 30 min of passive delivery (without Mg battery power), while RB penetrated the stratum corneum into the dermal area from the patch (Fig. 5d). RB delivered by an iontophoresis patch (with a $10^3 \Omega$ resistor) penetrated the skin with a thickness of ~ 2 times deeper than the VH20 hydrogel after 1 h. In conclusion, this Mg battery-powered iontophoresis patch is effective in delivering anionic drugs into and through the skin via the transdermal route.

Fig. 5 (d) Optical and fluorescent cross-sectional images of porcine skin via passive delivery and iontophoresis patch (RB was used as a model drug for easy observation; scale bar: 200 μm).

9. As the medical tape is gas permeable, will the hydrogel dry out in air?

Response: Thanks for the comments. We examined the water retention capability of the hydrogel in the patch covered with medical tape in a humid environment suitable for mice survival. The hydrogel remained at 80% water after 5 days of incubation at 25°C with a relative humidity of 70% (Fig. R5), indicating that the hydrogel will not dry out in the air during the treatment of psoriasis mouse.

Fig. R5 Water retention capability of VH20 hydrogel in the iontophoresis patch covered with a medical tape in the air.

10. 1) Page 12, line 303, how is the drug release efficiency 26% calculated? 2) The following statement cannot be obtained from Fig. 4g: “The release efficiency was approximately 26% when discharged with 1000 Ω installed resistors for 180 min, similar to 29% achieved for the potential controlled method at -0.6 V (Fig. 4g)”. The authors should refer to Table S2 or Fig. 4C.

Response: Thanks for the valuable suggestions.

1) The cumulative release efficiency is calculated according to Equation 5:

$$E_r = \frac{V_e \sum_{i=1}^{n-1} C_i + V_0 C_n}{m_{drug}} \times 100\%$$

where E_r is the cumulative release efficiency of the Dex, V_e is the displacement volume of PBS solution, V_0 is the total volume of release medium, C_i is the release fluid concentration of the replaced PBS solution, m_{drug} is the total drug-loading capacity, and n is the number of PBS replacements.

2) Thanks for pointing it out. We have used the wrong figure number. The description is revised on Page 11 as follows:

The release efficiency was approximately 25.8% when discharged with a 10³ Ω resistor for 180 min (Fig. 4f), similar to 29% achieved at an applied potential of -0.6 V (Fig. 4c).

11. Page 10 line 263, why would the drug release profile reach a plateau?

Response: Thanks for the comments. The process of electro-stimulated drug release is usually divided into a drug burst period and a sustained-release period. When a voltage was applied to the P(AM-co-SV) hydrogel, positively charged viologen (V^{2+}) was reduced to the neutral state (V^0), leading to a reduced binding force between the viologen and Dex drug. Therefore, a dumping effect was observed within the first 90 min due to the electrorepulsion force and high concentration gradient between the drug reservoir and the release medium, resulting in a fast rate of drug migration.

Subsequently, the drug migrations decrease as the free drug concentration in the hydrogel decreases because part of Dex is bound to the positively charged hydrogel backbone (V^{2+}). The drug release behavior from the hydrogel reaches a sustained release phase, thus approximating a plateau in the drug release profile (*Chem. Rev.* 2016, 116, 2602-2663).

12. Other minor issues: **1)** Fig. 1a, chemicals should be labeled with names; **2)** Fig. S5 a, the bar chart is not elucidative. Line chart is better; **3)** Page 10 line 255, and Fig S8 b, the type of LED that the biobattery can light up needs to be specified, as it indicates different level of output voltage.

Response: Thanks for your suggestion. We have revised the figures as suggestions.

1) Simplified chemical names are labeled in Fig. 1a.

2) A new line chart is provided in Fig. S5a.

3) The following description has been added to the figure caption of Fig. S8b.

Photographs of LED bulbs (3 mm, 2.2-2.4V) connected with VH20 hydrogels at distances of 0.5 cm, 1 cm, and 2 cm, respectively.

REVIEWERS' COMMENTS

Reviewer #1 (Remarks to the Author):

The iontophoretic acceleration of "release" and "penetration" have been still confused. The illustrated mechanism of the transdermal drug delivery in new Fig.5a is strange. "Electrorepulsion (electrostatic repulsion?)" is shielded by electrolyte ions. Dex anion will be released from the reduced VH20 hydrogel ($V^{++} \rightarrow V^0$) and supplied onto the skin surface. On the other hand, the transdermal Dex penetration is driven by "Electromigration" and "Electroosmosis" within the skin. Because the skin tissue is negatively charged (cationic permselective) due to immobile collagen, the dominant mechanism of skin penetration is the electromigration of cationic drug from anode, and/or the electroosmotic flow from anode to cathode. Therefore, the iontophoretic drug penetration from cathode is conceptually unattractive. Also, the paper cited in the response letter (Int. J. Pharm. 2021, 121009) explained that the efficiency of cathodal delivery of Dex is low due to the cationic perm-selectivity of skin (page 4 and page 6).

In new Fig.5b and 5c, cumulative amount of permeation was studied by changing the value of transdermal current. It was fortunately shown that the current application shows positive effect even at the cathode. However, the case of without current (VH20) is that without release. In addition, the permeation amounts for three current values (at 100, 1000, 10000 ohm) were not significantly changed even for 2-digit difference in current values. Acceleration of penetration by electric current at cathode is thus considered to be insignificant.

Authors re-emphasize the highlights of their work in the reply letter: (1) Drug release from biocompatible ion-conductive redox polymer, (2) Simple structure, (3) Built-in Mg battery. This is a nice paper of a biocompatible drug "release" patch and could be more suitable for a specialized journal about functional materials in health and medicine. In that case, it is needed to explain why a controlled on-skin release is required.

The development of the system of "anodic release" is highly expected.

Reviewer #2 (Remarks to the Author):

The authors addressed all my comments. Just one more suggestion for Table S1. The reported release capacity has different units, such as mol/h, mg/cm², ug/mL, %. It will assist comparison much easier if the authors can summarize them in a consistent unit, which might require some re-calculation of the reported values. The unit of the release capacity of ref 8 seems to be wrong, probably should be "35 $\mu\text{g cm}^{-2}$ " instead of "35 $\mu\text{g cm}^2$ ".

Reviewer #1:

1. 1) The iontophoretic acceleration of "release" and "penetration" have been still confused. **2)** The illustrated mechanism of the transdermal drug delivery in new Fig.5a is strange. "Electrorepulsion (electrostatic repulsion?)" is shielded by electrolyte ions. Dex anion will be released from the reduced VH20 hydrogel ($V^{++} \rightarrow V^0$) and supplied onto the skin surface. **3)** On the other hand, the transdermal Dex penetration is driven by "Electromigration" and "Electroosmosis" within the skin. Because the skin tissue is negatively charged (cationic permselective) due to immobile collagen, the dominant mechanism of skin penetration is the electromigration of cationic drug from anode, and/or the electroosmotic flow from anode to cathode. Therefore, the iontophoretic drug penetration from cathode is conceptually unattractive. Also, the paper cited in the response letter (Int. J. Pharm. 2021, 121009) explained that the efficiency of cathodal delivery of Dex is low due to the cationic perm-selectivity of skin (page 4 and page 6).

Response:

1) Thank you for your comment. The iontophoretic acceleration of drug delivery can be evidenced by the drug release amount and penetration rate. The cumulative drug release increased with the increase of output current provided by the embedded battery (Fig. 4f and 4g), which strongly supported the accelerated drug release using the patch. The accelerated drug penetration has been certified by 2.54 times ($10^2 \Omega$) higher penetration rate than the natural diffusion when applying the same amount of Dex on the porcine skin (Fig. R1, illustrated in the previous response letter).

Fig. R1 Cumulative deposition into the porcine skin from the iontophoresis patch (no-drug loaded hydrogel) with the same amounts of drugs on the skin.

2) Thanks for your comments. Electrorepulsion (also known as electromigration) and electroosmosis are two mechanisms involved in iontophoretic transport. Electrorepulsion is the primary mechanism for the iontophoretic transport of charged drugs (*Pharmaceutics* 2022, 14, 525). The drug-loaded viologen hydrogel electrode is reduced accompanied by the drug release, which is a typical phenomenon for conducting polymers to expel dopants in electrolytes when reduced at an applied current or potential. Similar mechanisms have been reported for the repulsion of dopants in conducting polymer films powered by biofuel cells.

(*J. Mater. Chem.* 2008, 18, 3608-3613; *Nano Energy* 2019, 66, 104120; *J. Am. Chem. Soc.* 2020, 142, 11602-11609; *Adv. Drug Deliver. Rev.* 2004, 56, 619-658).

3) Thanks for your comments. Dexamethasone sodium phosphate is a widely used anionic drug model for iontophoresis, with a negligible effect of electroosmotic flow on Dex transport efficiency. Delivery from the cathode (with electrorepulsion as the main driving force) is more appropriate than that from the anode (*Physical Therapy*, 2008, 1177-1185). In this work, the cathode efficiently transported Dex with a release amount of 2.69 mg g⁻¹ and permeation amount of 10.2 μg cm⁻² in 3 hours, which was effective in the treatment of psoriasis.

Furthermore, the paper cited in the previous response letter (*Int. J. Pharm.* 2021, 121009) supports the effective cathodal delivery of Dex, achieving therapeutically relevant concentrations after the current application for only 5 min. The slightly higher delivery efficiency of buflomedil hydrochloride (cationic drug) from the anode than the cathodic delivery of Dex (anionic drug) may be caused by the different drug structures and cationic permeability of the skin.

2. In new Fig.5b and 5c, cumulative amount of permeation was studied by changing the value of transdermal current. It was fortunately shown that the current application shows positive effect even at the cathode. 1) However, the case of without current (VH20) is that without release. 2) In addition, the permeation amounts for three current values (at 100, 1000, 10000 ohm) were not significantly changed even for 2-digit difference in current values. Acceleration of penetration by electric current at cathode is thus considered to be insignificant.

Response:

Thanks for your comments.

1) There did exist natural drug release behavior due to the presence of free drugs in the hydrogel. Drug release from an iontophoresis patch powered by a Mg battery includes current-controlled release and natural diffusion. Natural drug release without the current stimulation was 0.52 mg g⁻¹ (Fig. 4f).

2) The drug permeation amount (Fig. 5b) was greatly enhanced when applying a large transdermal current, including 4.43 times (10² Ω: 1.19 mA cm⁻²), 3.21 times (10³ Ω: 0.18 mA cm⁻²), and 3.10 times (10⁴ Ω: 0.03 mA cm⁻²) higher than that without battery power. These results strongly support the accelerated penetration at the cathode, and the cathodic delivery is effective. Regarding the non-linear relationship between the applied current and the permeation amount, it may be explained by the fact that the anionic Dex was effectively penetrated into the skin at the cathode in response to small transdermal currents, as well as the contribution from the natural diffusion.

3. Authors re-emphasize the highlights of their work in the reply letter: (1) Drug release from biocompatible ion-conductive redox polymer, (2) Simple structure, (3) Built-in Mg battery. This is a nice paper of a biocompatible drug "release" patch and could be more suitable for a specialized journal about functional materials in health and medicine.

In that case, it is needed to explain why a controlled on-skin release is required.
The development of the system of "anodic release" is highly expected.

Response:

Thanks for your comments. However, we believe that our work fits in the scope of *Nature Communications*. Wearable transdermal iontophoresis offers advantages for patient-comfort when deploying epidermal disease treatments. It may attract a broad readership from the field of material chemistry, pharmaceuticals, and biomedical engineering.

The ionic conductive hydrogel as the integrated drug-carrying cathode provides an on-demand cathodic drug release and penetration via the transdermal route. Future work may focus on using conductive hydrogel as the integrated anode to deliver drugs.

Reviewer #2:

The authors addressed all my comments. **1)** Just one more suggestion for Table S1. The reported release capacity has different units, such as mol/h, mg/cm², µg/mL, %. It will assist comparison much easier if the authors can summarize them in a consistent unit, which might require some re-calculation of the reported values. **2)** The unit of the release capacity of ref 8 seems to be wrong, probably should be "35 µg cm⁻²" instead of "35 µg cm²"

Response:

1) Thanks for your suggestion. We have revised Supplementary Table 1 as suggested (please see below). The reported drug release capacities have been re-calculated and shown in µg cm⁻².

Supplementary Table 1 Summary of current iontophoresis systems.

Category	Typical materials	Output	Model drug	Release capacity (µg cm ⁻²)	Application	Ref.
DC	—	0.3mA	Dex	46.15	The objective was to study the competition of chloride released from a Ag/AgCl cathode on the iontophoretic delivery of dexamethasone phosphate	1
	Microneedle array	0.5mA/1mA	Insulin nanovesicles	—	A transdermal patch that combines microneedle array (MA) with iontophoresis can achieve synergistic and remarkable enhancement of drug delivery with precise electronic control.	2
	Poly(ethylene glycol) hydrogel	100 mA/cm ²	Dextran-40k/ Bevacizumab	1443 / 900	An iontophoresis device based on a hydrogel ionic circuit (HIC), for high-efficiency intraocular macromolecule and NP delivery are described.	3
	Polypyrrole nanoparticles	0.13 mA/cm ²	Insulin	68.29	This work aims to investigate the potential of water-soluble polypyrrole nanoparticles as a drug donor matrix for controlled transdermal iontophoresis of insulin.	4

PENG	Poly(lactic acid-gold-PPy microneedles	100 V / 2 μ A	Dex	4.3	Collect and convert biomechanical energy from human movement into electrical energy to control drug release on-demand and present in psoriasis treatment.	5
TENG	DOX loaded red blood cells on the Cu film	70 V / 0.5 μ A	DOX	—	Killing cancer cells both in vitro and in vivo at a low drug dosage.	6
	Poly(3-hexylthiophene) films	647 V / 165 μ A	Salicylic acid	—	TENG can provide a steady voltage supply for sustainable drug release.	7
	PPy/Dex film on gold electrode	100 V 19 mW/cm ²	Dex	35	Electricity generated from TENG was used to power iontophoresis treatment.	8
	Silicon nanoneedles-array	20 V / 4 μ A under 1rps	siRNA dextran-FITC	688	TENG-driven electroporation system is developed for intracellular drug delivery in vivo and in vitro.	9
	Poloxamer hydrogel	1200 V / 20 μ A	Rhodamine 6G methylene blue	2.63 / 11.3	A wearable TENG is used as the motion sensor and energy harvester for iontophoresis without stored-energy power sources.	10
	Polydimethylsiloxane drug reservoir	15 V / 1.5 mA	Fluorescent microparticles	—	TENG-based self-powered implantable drug-delivery system demonstrates its functionality for ocular drug delivery.	11
Biofuel cell	Biocathode: PEDOT functionalized gold electrode Bioanode: Carbon nanotubes	0.4 V 33.8 mW/cm ²	Acetaminophen	—	A biocomputing, logic-based detection method with a controlled-release drug delivery actuator that is based on a closed-loop, self-powered enzymatic biofuel cell system.	12

	Biocathode: PEDOT doped onto the NPG/Os(bpy)₂ PVI-BOx Bioanode: NPG/Os-(bpy)₂PVI-GOx	0.377 V 1.35 μW/cm²	Ibuprofen Fluorescein 4',6-diamidino-2-phenylindole	197 /0.101/ 0.102	A self-powered, controlled drug-release system based on bilayer-modified electrodes has been demonstrated. Three model compounds bearing negative or positive charges were released in a controlled manner.	13
	Biocathode: BOD-modified O₂-diffusion carbon fabrics Bioanode: FDH-modified carbon fabrics	0.55-0.7 V 10-50 μA/cm²	Rhodamine B Ascorbyl glucoside	—	BFC-driven current-assisted penetration of ascorbyl glucoside and rhodamine B into the skin.	14
	Biocathode: PSP/carbon sphere /glassy carbon Bioanode: PDS/gold nanobowl/ glassy carbon	145 ± 1.2 μW/cm²	DOX	1.15	A robust glucose/O₂ fuel cell-based biosensor is successfully integrated with a targeted drug delivery system to create a self-sustained and highly compact drug delivery model with self-diagnosis and self-evaluation.	15
	Biocathode: BOD-immobilized electrode Bioanode: FADGDH complex immobilized electrode	0.407 V 157 μW/cm²	—	—	An autonomous, self-powered, sensing actuator that employs the principle of bio-capacitor as the core technology.	16
Galvanic cell	Cathode: PEDOT Anode: Zn	—	Sulforhodamine B	—	A battery-driven drug delivery device powered by physiological pH can target intended sites and be actuated galvanically to trigger localized drug release.	17

	Cathode: PPy Anode: Mg	—	Adenosine triphosphate (ATP)	—	The drug release from this system can be performed without the external power source, and the massive drug release only occurs at around human body temperature.	18
	Cathode: AgCl Anode: Zn Drug loading material: PEDOT:PSS/PA Am hydrogel	1.08 V / 0.2 mA 60 μ W	Rhodamine B	—	Iontophoretic drug delivery device, and an electrical nerve stimulator, for transcutaneous bioanalysis and therapy.	19
	Cathode: PEDOT Anode: Zn	—	Gold nanoparticles	—	In vivo study of PEDOT/Zn artificial micromotors using an in vivo mouse model and characterization of their distribution, retention, and toxicity.	20
	Cathode: P(AM-co-SV) hydrogel Anode: Mg	1.1 V/10 mA cm^{-2} 3.57 mWh cm^{-2}	Dex	139.6	Biobattery-powered iontophoresis generated an on-demand transdermal release profile without complicated feedback circuits due to its straightforward potential-controlled release mechanism.	This work

Notes

DC: direct current; PENG: piezoelectric nanogenerator; TENG: triboelectric nanogenerator; BFC: enzymatic biofuel cell; NPG/Os(bpy)₂PVI-BOx: Os(bpy)₂PVI-mediated bilirubin oxidase (BOx);

NPG/Os-(bpy)₂PVI-GOx: [Os(2,2'-bipyridine)₂(polyvinylimidazole)₁₀Cl]^{+ /2+} (Os(bpy)₂PVI)-mediated glucose oxidase (GOx); BOD: bilirubin oxidase; FDH: fructose dehydrogenase; PSp: phosphatidylserinebinding peptide; PDS: partial complementary DNA double strand-decorated; FADGDH: flavin adenine dinucleotide-dependent glucose dehydrogenase; —: not provided.

2) Thank you for pointing out our mistake. The unit of the release capacity of ref 8 is $\mu\text{g cm}^{-2}$, which has been revised in Supplementary Table 1.